# The Vulnerability, Impacts, Adaptation and Climate Services Advisory Board (VIACS AB v1.0) contribution to CMIP6

Alex C. Ruane[1], Claas Teichmann[2], Nigel W. Arnell[3], Timothy R. Carter[4], Kristie L Ebi[5], Katja Frieler[6], Clare M. Goodess[7], Bruce Hewitson[8], Radley Horton[9], R. Sari Kovats[10], Heike K. Lotze[11], Linda O. Mearns[12], Antonio Navarra[13], Dennis S. Ojima[14], Keywan Riahi[15], Cynthia Rosenzweig[1], Matthias Themessl[16], and Katharine Vincent[17]

[1]NASA Goddard Institute for Space Studies, New York, USA
[2]Climate Service Center Germany (GERICS), Helmholtz-Zentrum Geesthacht, Hamburg, Germany
[3]Department of Meteorology, University of Reading, UK
[4] Finnish Environment Institute (SYKE), Helsinki, Finland
[5]Department of Global Health, University of Washington, Seattle WA, USA
[6]Potsdam Institute for Climate Impact Research (PIK), Potsdam, Germany
[7]Climatic Research Unit, School of Environmental Sciences, University of East Anglia, UK
[8]Climate System Analysis Group, University of Cape Town, South Africa
[9]Center for Climate Systems Research, Columbia University
[10]Centre on Global Change and Health, London School of Hygiene & Tropical Medicine, UK
[11]Department of Biology, Dalhousie University, Halifax, Nova Scotia, Canada
[12]National Center for Atmospheric Research, Boulder, USA
[13]Centro EuroMediterraneo sui Cambiamenti Climatici e Istituto Nazionale di Geofisica e Vulcanologia, Bologna, Italy
[14]Department of Ecosystem Science and Sustainability, Colorado State University, Fort Collins, USA
[15]International Institute for Applied Systems Analysis, Laxenburg, Austria
[16]Climate Change Centre Austria (CCCA), Vienna, Austria
[17]Kulima Integrated Development Solutions (Pty) Ltd and University of the Witwatersrand, Johannesburg, South Africa

*Correspondence to:* Alex Ruane (alexander.c.ruane@nasa.gov)

**Abstract.** This paper describes the motivation for the creation of the Vulnerability, Impacts, Adaptation and Climate Services (VIACS) Advisory Board for the Sixth Phase of the Coupled Model Intercomparison Project (CMIP6), its initial activities, and its plans to serve as a bridge between climate change applications experts and climate modelers. The climate change application community comprises researchers and other specialists who use climate information (alongside socioeconomic and other environmental information) to analyze vulnerability, impacts and adaptation of natural systems and society in relation to past, ongoing and projected future climate change. Much of this activity is directed toward the co-development of information needed by decision-makers for managing projected risks. CMIP6 provides a unique opportunity to facilitate a two-way dialogue between climate modelers and VIACS experts who are looking to apply CMIP6 results for a wide array of research and climate services objectives. The VIACS Advisory Board convenes leaders of major impact sectors, international programs, and climate services to solicit community feedback that increases applications relevance of the CMIP6-Endorsed Model Intercomparison Projects (MIPs). As an illustration of its potential, the VIACS community provided CMIP6 leadership with a list of prioritized climate model variables and MIP experiments of greatest interest to the climate model applications community, indicating the applicability and societal relevance of climate model simulation outputs. The VIACS Advisory Board also recommended an impacts version of Obs4MIPs, and indicated user needs for the gridding and processing of model output.

## 1 Introduction

Charles David Keeling's observations of rising carbon dioxide concentrations at the Mauna Loa Observatory alerted the world to the formidable challenge of anthropogenic interference in the climate system more than 50 years ago (Keeling, 1960). In the years since there has been tremendous progress in our understanding of climate drivers, atmospheric circulation, interaction between climate system components, climate dynamics, human and natural system responses to climate change, and strategies that may safeguard these systems in a changing world (IPCC, 2013). The collective evidence base compiled by the climate science community culminated in action by the United Nations Framework Convention on Climate Change (UNFCCC) to adopt the 2015 Paris Agreement to limit warming of the global climate and to increase the ability to adapt to adverse climate impacts (UNFCCC, 2015). The Paris Agreement reinforces the urgent need for climate applications based on cutting-edge science to support the implementation of emissions reductions and climate adaptations around the world while not undermining social well-being. It is therefore crucial that a platform is created to support an active dialogue between researchers and practitioners so that information exchange about climate change, sectoral system responses, and strategies to respond can be sustained.

Climate research is based on a foundation of observational data and understanding of the physical, chemical, and biological processes that govern the climate system. Climate models, bolstered by an exponential increase in computational resources, have emerged as an important tool for climate scientists seeking to fill gaps in knowledge of the climate system. In particular, climate models play an important role in simulating complex and interacting climate processes, testing climate hypotheses, illustrating the potential ramifications of emissions pathways, and acting as a virtual laboratory of climate response. The Coupled Model Intercomparison Project (CMIP) emerged out of the earlier Atmospheric Model Intercomparison Project (AMIP – Gates et al., 1999), recognizing the rapid development from atmosphere-only general circulation models (GCMs) toward coupled ocean–atmosphere–cryosphere–land GCMs. The establishment of CMIP in 1995 was seen as an initiative to undertake systematic intercomparison and evaluation of climate models to spur model improvement and application of comparable outputs (Meehl et al. 2000).

The range of expertise required to develop climate models differs in many respects from the expertise underpinning studies of climate change vulnerability, impacts and adaptation (VIA). Although there are many overlapping areas of inquiry (e.g., vegetative response is of interest in climate models, for agricultural and forestry applications, and in ecosystem science), VIA experts commonly translate the physical quantities reported in climate output (e.g., temperature, precipitation, humidity) into societally-relevant quantities (e.g., crop and fisheries yield, available water and energy resources, disease prevalence, commodity market shifts, or species habitat loss). However, this translation process frequently demands much more than a deterministic representation of a climatic "cause" producing an "effect" on a given exposed system. System response under a changing climate is frequently mediated by parallel societal and environmental ("global") changes (Revi et al. 2014). It can also be influenced by factors that may be poorly understood and difficult to model (e.g., aspects of behavior, vulnerability, and governance) that require other expertise and methods to be deployed. Some VIA analysis therefore takes a 'bottom-up' approach starting from a consideration of

the factors affecting vulnerability to impact, rather than a 'top-down' scenario-driven approach, and in such analyses information on potential climate changes may play only a small role. Hence, the science of VIA analysis is both interdisciplinary and demands extensive knowledge of climate, other concurrent global changes (biophysical and social), and the affected system itself (Adger et al. 2013).

VIA analysis is undertaken in varying contexts, ranging from publicly-funded academic research (e.g. developing new paradigms, methods, datasets or tools) to applications delivering products directly to specific clients with particular geographical areas or sectors of concern. The realm of climate services (CS, see below) is a subset of the latter category, in which experts combine sector-specific climate and impacts information to form knowledge products and tools for decision support across public and private stakeholders. This "operationalizing" of climate science requires an understanding of decision-making needs, processes, timelines, incentives, priorities, level of risk-aversion, and tradeoffs that determine the tailored climate information products that would be most useful, for example (Weaver et al. 2014). This understanding can, in turn, inform VIA methods, tools, and data products, particularly on inter-and transdisciplinary frontiers. Figure 1 provides a simplified schematic of the interactions between the science of climate, the science of system behavior, and the operationalization of climate information.

This paper describes the origins, motivation, creation, and initial activities of the Vulnerability, Impacts, Adaptation and Climate Services (VIACS) Advisory Board for CMIP, which is designed to facilitate communications between the climate modeling community and the various communities applying climate change information for scientific or operational purposes. By formalizing this process and involving leaders from each community, the VIACS Advisory Board aims to enhance the societal benefit of climate information.

## 2 Background
### 2.1 CMIP6

After its founding in 1995, the Coupled Model Intercomparison Project (CMIP) timed its phases to provide climate model projections of record for the Intergovernmental Panel on Climate Change (IPCC) Assessment Reports (AR). CMIP2, CMIP3, and CMIP5 formed the basis of global model simulations for the Third Assessment Report (TAR), Fourth Assessment Report (AR4), and Fifth Assessment Report (AR5; IPCC, 2015), respectively. CMIP is now in its sixth phase (CMIP6; Eyring et al., 2016a), and continues in its role of systematically inter-comparing climate models and making outputs available to the applications communities in support of all three Working Groups of the Sixth IPCC Assessment Report cycle (AR6).

CMIP6 is designed to answer three overarching science questions (Eyring et al., 2016a): (1) How does the Earth System respond to forcing? (2) What are the origins and consequences of systematic model biases? and (3) How can we assess future climate changes given climate variability, predictability and uncertainties in scenarios? CMIP6 is organized around a historical climate simulation, entry card simulations for CMIP6 designed for Diagnostic, Evaluation and Characterization of Climate (or "Klima" in Greek, giving an acronym DECK for these central

simulations), and a number of CMIP6-Endorsed Model Intercomparison Projects (MIPs) that explore specific aspects
of climate, model performance, and/or diagnostics (Table 1). CMIP6-Endorsed Diagnostic MIPs are unique in that
they do not define individual model experiments, but commit to specific aspects of analysis and contribute to
evaluation and application. These central experiments and CMIP6-Endorsed MIPs were designed within the scientific
backdrop of the World Climate Research Programme's Grand Science Challenges (see Eyring et al., 2016a). CMIP6
provides participating modeling groups with an overarching structure, coordination, data framework, and hub to
communicate results to the broader community, potentially including online visualizations and analyses.

### 2.2 Applied Climate Communities

Observations and understanding of the effects of climate and weather on natural and human systems have raised
concerns about potential adverse impacts of anthropogenic climate change, and about decisions that may be required
for preparing and adapting systems to these impacts. Such concerns have motivated the development of practical
approaches for analyzing impacts, making use of model projections of future climate along with scenarios describing
concurrent changes in socioeconomic conditions affecting system exposure and vulnerability.

### 2.2.1 The Vulnerability, Impacts, and Adaptation (VIA) research community

In a review for the IPCC AR5, Burkett et al. (2013) documented the emergence and rapid increase in climate impacts
research, beginning with agricultural and biological research in the 1970s and then expanding into many areas of
social science. To illustrate this evolution, they report that more than 100 papers were published on the topic of climate
change "impacts" in 1991, with the topics of "adaptation" and societal "cost" only reaching that threshold in 2003.
VIA publications still come disproportionately from European, North American, and Asian-Pacific institutions and
focus largely on impacts in those regions, however VIA publications from other regions became more numerous in
recent years.

The evolution of VIA literature is also evident in successive assessments by IPCC Working Group II (IPCC, 1990,
1992, 1996, 1997, 2001, 2007, 2014). The organization of the assessments evolved with the development of the subject
area, from largely impacts-orientated chapters in the first three full assessments (IPCC, 1990, 1996, 2001) toward a
greater focus on adaptation and risk management across the working group in the latest two assessments (IPCC, 2007,
2014). All assessments employed a sectoral and thematic treatment of VIA issues, with additional regional chapters
introduced following the Second Assessment (IPCC, 1997). The majority of the literature was based on studies with
a local-to regional-scale focus, though there are also studies examining global impact or using integrated assessment
models. Very few studies use systematic methods across sectors taking a global perspective (e.g., Arnell, 2016;
Warszawski et al., 2014). One of the challenges faced in Working Group (WG) II has been the need to aggregate and
synthesize across multiple studies, sectors and regions, to identify key risks of climate change to be communicated to
decision makers.

The researchers and practitioners conducting VIA studies are spread across many thousands of institutions, worldwide, with very few centers dedicated to VIA research. Until the establishment of PROVIA in 2010 (see below), there has been no single international program coordinating a research agenda to which most VIA researchers would naturally be aligned (equivalent to the World Climate Research Programme for climate researchers or the Integrated Assessment Modeling Consortium for mitigation researchers). The IPCC assessments have been among the few examples wherein

hundreds of senior VIA researchers come together to review and interpret the latest published research findings within a coherent framework. In this connection, there have been calls for consistency in approaches to VIA studies, to facilitate more effective comparison and integration of results between studies and regions. The need was raised in methodological guidelines for impact and adaptation assessment developed by the IPCC ahead of the first UNFCCC Conference of the Parties (IPCC, 1994b). Moreover, one of the original motivations for establishing the IPCC Task

Group on Scenarios for Climate Impact Assessment (TGCIA) in 1997, the forerunner of TGICA (see section 4.1.1, below), was to help encourage the selection and application of a consistent set of climate and socioeconomic scenarios in climate change impact and adaptation studies (Parry, 2000). Ten years later, Rosenzweig and Wilbanks (2010) called for systematic intercomparison and evaluation across VIA methods and scales, as well as self-organization to increase communication within the community and with collaborators in the climate modeling and integrated

assessment modeling communities. Nascent efforts to build cohesively organized research endeavors within various impact sectors and international programs provide a framework for VIA interaction with CMIP6 (as described in Section 4).

**2.2.2 The Climate Services community**

Climate services seek to enhance stakeholders' abilities to anticipate and build resilience to changing climate conditions through the co-design and co-production of tailored information for climate product development and user application. Such activities themselves are probably as old as climate research. However, it is only in recent years that the term "climate services" has come into widespread usage. There are several recent definitions of "climate services" emphasizing different aspects (Laurenco et al., 2016). The World Meteorological Organization's (WMO) Global

Framework for Climate Services (GFCS; WMO, 2014) and the American Meteorological Society's (AMS) definitions focus on the aspect of the preparation and delivery of user-tailored climate data. The definition in the Climate Service Roadmap, a European Commission initiative to foster research and innovation for climate services, also includes "counselling on best practices, development and evaluation of solutions and any other service in relation to climate that may be of use for the society at large" (European Commission, 2015).


    A brief history of climate services is provided by Vaughan and Dessai (2014). They localize the foundation of climate services to the International Meteorological Organization (IMO; a precursor to the WMO) in the late 19th century. The World Climate Programme was created in the context of the first World Climate Conference (WCC) organized by the WMO, aiming to improve our understanding of the climate system and its impact on society. More recently the

GFCS was created by the WMO to provide a worldwide mechanism for coordinated actions to enhance the quality, quantity and application of climate services (WMO, 2014). An open, informal international coalition was founded in

the frame of the first international conference on climate services (ICCS 1) in New York, 2011: the Climate Services Partnership. It aims at improving the provision and development of climate services worldwide and at supporting the GFCS. Growing interest in climate services recognizes the fact that, despite the rapid improvement and growth in the information base for understanding past climate events and future projections, much of this information is not informing climate risk management (McGregor, 2015; Eisenack et al., 2014). This also reflects the growing awareness that Climate Services have specific characteristics that may differentiate them from the established Meteorological Forecast Services; including the multidisciplinary nature of the information required and the innovative climate service co-design process.

## 3 The VIACS Advisory Board

### 3.1 Motivation

The need for strong communication and collaboration between the climate modeling community and those who apply climate information has long been recognized, as there is a common need to:

- keep climate applications up to date on the latest model developments, outputs, and evaluations;
- track the ways in which climate model simulations inform the identification and prioritization of risk management and resilience-building strategies;
- evaluate the effectiveness of climate services;
- provide feedback into priority areas for model improvements;
- define variables for the CMIP6 data request that are relevant for the VIACS community; and
- advise applications communities that do not have access to the technical skills and/or resources necessary to interpret CMIP model archives.

In the past these lines of communication have been formed in an *ad hoc* fashion that too often lacks stability or falls well short of its potential.

Figure 2a presents an illustration of the lines of communications (gray lines) between climate modeling centers (black stars) and various VIACS communities (represented as colored shapes of various sizes and types). Although many lines of communication have been forged over the years, their utility varies widely. These include formal relationships or memoranda of understanding at center levels, national projections services that coordinate with VIACS communities (but not back to CMIP), co-located climate modeling and VIACS groups, VIACS communities that have made strong efforts to reach out to many climate modeling centers (or vice versa), strong connections between individual modeling centers and individuals within a VIACS project, lines of communication developed for a particular project, and some groups that remain isolated with few lines of communication. Soliciting the VIACS perspective for climate modeling or climate model center perspectives on VIACS applications has been an onerous and complex task involving many actors and organizations.

Figure 2b illustrates the potential for the VIACS Advisory Board for CMIP to play an *additional* role in communication between the climate modeling centers and VIACS communities. Utilizing CMIP's ability to organize and act as a communications hub for the modeling centers, the VIACS Advisory Board is similarly designed to survey the leaders of major VIA sector disciplines (e.g., agriculture, water resources, forestry, fisheries, terrestrial and marine ecosystems, infrastructure, urban, health, energy), regional integrated impacts studies, international agencies and committees, and projects (examples are described in Section 4 below). These leaders are often well-connected with the broader VIACS communities in their same field, allowing a manageable group of contacts to provide more coherent access to the broader VIACS communities. Depending on the request, information may be requested by discipline, project, or specific region, which allows solicitations to be efficiently targeted.

### 3.2 Endorsement, Mandate, and Formation of the VIACS Advisory Board

To form a more coherent and productive interaction between the climate modelers in CMIP6 and the VIACS communities, and to enhance the relevance of CMIP6 to society through all impact sectors, CMIP6 endorsed the creation of a VIACS Advisory Board for CMIP6. Launched in 2015 as a Diagnostic Model Intercomparison Project (MIP), the VIACS Advisory Board was not proposed to conduct new climate model experiments, but serves as an advisory body to encourage inputs from the VIACS community on experiment and output design for CMIP6-Endorsed MIPs, guidelines for good practices in the use of CMIP6 outputs, and online metrics and visualizations intended for use by the VIACS community. The VIACS Advisory Board is designed to be a bridge between the VIACS community (generally those researchers whose work is assessed by IPCC Working Group II – *Impacts, Adaptation, and Vulnerability*) and the climate modeling community (generally those researchers whose work is assessed by IPCC Working Group I – *The Physical Science Basis*). Climate modeling groups that are interested in building stronger engagement with the climate change applications community, and likewise VIACS experts eager to spur climate model developments that would facilitate applications, are encouraged to interact with the VIACS Advisory Board.

Engagement with the CMIP modeling groups will help ensure that model output fits the climate service application needs, and also allows the modeling groups to provide synthesized input into the process by which climate information is distilled into climate applications messages. A close connection is also needed to CORDEX (also a CMIP6 Diagnostic MIP, see Section 4.1.4 below) to motivate downscaling methods geared towards providing improved climate information on temporal and spatial scales required in applications research and climate services, as well as to TGICA (see Section 4.1.1 below) to ensure consistency in scenarios for climate applications. Both groups also contribute valuable experience working in the climate modeling and climate applications communities. The VIACS Advisory Board will advise on the establishment of common evaluation concepts for global and regional climate data, best practices for the creation of individual climate service products, and online visualizations developed by CMIP to explore the sectoral implications of climate projections. Another goal of the Board is to help improve the ways that climate services present information (e.g., vocabulary, uncertainties, information content, product consistency, the delivery and perception of messages). This can benefit from social science networks within the VIACS community.

### 3.3 Structure

The VIACS Advisory Board is led by Co-Chairs; one each from the VIA and the Climate Services communities (initial co-Chairs were leaders of VIA and CS proposals combined by the CMIP Panel). Board members serve two-year terms with rotating chairs to ensure new perspectives and a reasonable time commitment. Members of the VIACS Advisory Board have a mandate to coordinate with other experts within their region/sector/group to provide community-based guidance that can be integrated at the VIACS Advisory Board level and then presented to CMIP6. Board members were selected by the co-Chairs and drawn from leaders of VIA sectors, major projects, and international programs, many having participated in several parallel engagement efforts that were merged into the original proposal for a VIACS Advisory Board within CMIP. Members are tasked with surveying their respective communities (not just their own inner circle) and providing comprehensive feedback for CMIP6 to consider in designing and prioritizing scenarios and metrics for analysis and benchmarking that would be relevant for VIACS applications. Future terms of the Advisory Board would benefit from the inclusion of more members from regions beyond North America, Europe, and South Africa; at this point membership reflects these regions' disproportionate role in leading international VIACS programs. It is worth noting that current Board members work beyond their home regions, so perspective and information needs of other regions are not entirely neglected. Board members also provide guidance from their experience developing metrics and visualizations that appeal to VIACS community researchers, stakeholders, and decision-makers. These include sector-specific indices (e.g., heat damage degree days for ecosystems, consecutive dry days for agriculture and water resources; temperature-humidity indices for health) and requirements for documentation and online guidance that will facilitate understanding of CMIP6 products by the lay public. The Board will also advise on the translation and dissemination of CMIP climate modelers' advice for best practices for the use of climate model outputs within the VIACS community.

### 3.4 Convening and Communications Plan

To fulfill its potential as a conduit for communication between the VIACS and climate modeling communities, the Board establishes regular communication between representatives of the CMIP6-Endorsed MIPs and the VIACS community. High-level participation from both sides is required. Each consultation of the VIACS Advisory Board comprises five steps (summarized in Table 2). The VIACS Advisory Board is expected to convene approximately on a quarterly basis; however in the early stages of CMIP6 the Board's activities have been closer to a monthly schedule in response to urgent CMIP6 design questions.

The VIACS Advisory Board is also active in periods between teleconferences. Activities include outreach encouraging greater utilization of the VIACS Advisory Board as a unique resource for both climate modelers and VIACS communities, as well as the development of new network connections that will increase CMIP's reach into the climate applications community. Representatives of the VIACS Advisory Board also participate in major CMIP6 meetings to give voice to the VIACS perspective on priority climate model outputs and evolving VIACS community needs, although any formal recommendations must be made in consultation with the full Advisory Board. Although the Board is tasked with providing feedback and ideas regarding the use of CMIP6 outputs for VIACS assessments, the

assessments themselves are beyond the mandate of the VIACS Advisory Board itself but are likely to involve many of the Board members through their participation in independent studies.

## 4 Engaging the broader VIACS communities

The VIACS Advisory Board is a focused effort specifically mandated to link the VIACS and GCM communities for CMIP6. A portion of this mandate is shared by a range of other groups, and the VIACS Advisory Board seeks to complement these efforts by offering an additional level of coordination and engagement among leaders. This section highlights a non-exhaustive selection of the major groups within various VIACS communities with whom the VIACS Advisory Board engages to solicit feedback and inputs for the CMIP process (for example in the course of step 4 of the VIACS consultation process summarized in Table 2).

### 4.1 International Programs

The VIACS Advisory Board builds on a legacy of research and applications networks and materials established by several high-profile expert groups and programs.

### 4.1.1 TGICA

Up to the time of the IPCC Second Assessment, while there was some coordination in the selection of scenarios describing alternative future developments of atmospheric greenhouse gas and aerosol emissions under the auspices of the IPCC (e.g. Leggett et al., 1992; IPCC 1994a), the consistent use of emissions scenarios as inputs to fully coupled AOGCMs run in transient (time-dependent) mode was still limited. Many GCMs were still being run for scenarios of doubling or quadrupling of $CO_2$; sensitivity-based simulation designs that were not suitable for many VIACS applications. Moreover, access to the outputs of climate model simulations had to be negotiated with the modeling centers themselves or through a few volunteer individuals and organizations who collected climate model information on behalf of a growing research community studying impacts (e.g. at the National Center for Atmospheric Research in the US and the Climatic Research Unit in the UK).

Ahead of the IPCC Third Assessment there was clear recognition of a need to engage and coordinate between different research communities whose work was based on the use of socioeconomic and greenhouse gas emissions scenarios. This resulted in the 1997 establishment of a Task Group on Scenarios for Climate Impact Assessment (TGCIA) to inventory impact studies and climate model runs, provide climate model outputs through a Data Distribution Centre (DDC; http://www.ipcc-data.org), and produce guidance materials to facilitate the use of scenarios. TGCIA and the DDC worked to facilitate cooperation and communication between the modeling and impacts communities, particularly with respect to the availability and accessibility of climate data. It was out of criteria suggested by TGCIA – for climate model simulations and the selection of standard variable datasets for downloading and storage – that the foundations for activities now coordinated by CMIP originated.

The IPCC Task Group on Data and Scenario Support for Impact and Climate Analysis (TGICA) is the present-day counterpart of TGCIA. It comprises members drawn from nominations by national IPCC Focal Points, bringing together diverse expertise and experiences from a cross section of research communities representing all three IPCC Working Groups. TGICA's current mandate is to "facilitate wide availability of climate change related data and scenarios to enable research and sharing of information across the IPCC Working Groups". TGICA maintains the DDC as a means of accessing climate, socio-economic and environmental data, both from historical observations and from future projections (scenarios), in support of IPCC work and as used in the IPCC assessments. The DDC is designed primarily for climate change researchers, but is also relevant to educators, practitioners, governmental and non-governmental organisations, and the public. Importantly, the DDC hosts data relevant across Working Groups with a consistent quality control and appropriate supporting materials.

TGICA also contributes to building capacity, for example by publishing several peer-reviewed technical guidelines, distributed by the DDC, on the development and application of climate, other environmental and socioeconomic scenarios for climate change impact, adaptation and vulnerability assessment (e.g. IPCC-TGICA 2007; Mearns et al., 2003; Nicholls et al., 2011; Wilby et al., 2004), with other similar documents and updates in preparation. In addition, TGICA facilitates expert meetings to contribute to regional capacity building. For example, an expert meeting on "Integrating analysis of regional climate change and response options" was held in 2007 to catalyse regional interdisciplinary research on climate change, impacts, adaptation, vulnerability and mitigation (Marengo et al., 2009).

### 4.1.2 PROVIA

The Global Programme of Research on Climate Change Vulnerability, Impacts and Adaptation (PROVIA; UNEP, 2013) represents an interface between the research community and decision-makers and other stakeholders to provide direction, coherence, and capacity-building at the international level for improved policy-relevant research on vulnerability, impacts and adaptation. PROVIA is recognized within the World Climate Programme as the body that helps represent the perspectives of this highly diverse, transdisciplinary community, operating for researchers associated with IPCC Working Group II in a manner similar to the World Climate Research Program (WCRP) coordination of research associated with Working Group I. PROVIA's parent organizations are the UN Environment Program (UNEP), the World Meteorological Organization (WMO), and the UN Educational, Scientific, and Cultural Organization (UNESCO). PROVIA helps international communities share practical experiences and research findings by improving the availability and accessibility of knowledge to the people that need it most. Together with collaborative partners, knowledge networks, and the larger VIACS community, it is helping to identify and alert international organizations to research needs and gaps. In this way PROVIA helps the scientific community to mobilize and communicate the growing basis of information from VIACS research so that governments and other key stakeholders are able to consider this knowledge in their decision making processes.

The VIACS Advisory Board was endorsed by the Programme of Research on Climate Change Vulnerability, Impacts, and Adaptation (PROVIA), which will act as an anchor program to support the long-term balance and stability of the

Advisory Board as well as to encourage participation of representatives from numerous regions, impacts sectors, and prominent international groups. PROVIA is focused on four objectives, each of which may be furthered by the VIACS Advisory Board: 1) Coordinating research on climate vulnerability, impacts, and adaptation; 2) Guiding investment in research; 3) Communicating high-quality scientific information to governments and international agencies with due urgency and specificity; and 4) building research capacity, especially in developing countries.  Specific PROVIA

activities of direct relevance to VIACS include co-sponsoring the biannual Climate Adaptation Futures Conference, developing a research agenda and guidance documents to support VIA assessment, supporting scenario development and model intercomparison activities, conducting VIA related training workshops, and supporting a fellowship program for young researchers. All these activities offer mechanisms for the VIACS Advisory Board to engage with a large number of researchers, stakeholders, decision-makers, and policy-makers to better integrate climate

information into climate change risk assessments across a number of sectors, with results also feeding back into the design and implications of climate modeling experiments.

### 4.1.3 The WCRP Working Group on Regional Climate

The Working Group on Regional Climate (WGRC) was established by the WCRP in 2013 with a mandate to

"coordinate regional climate research and science-based knowledge development for decision makers". This mandate to interact with both the physical climate science community (particularly within WCRP) and providers and users of climate information is reflected in the membership, terms of reference, and activities undertaken by the WGRC. For example, it has a specific role to oversee and promote CORDEX (see below) and in this context the emphasis has been on facilitating and guiding the tailoring and application of CORDEX outputs within regions (such as Latin America

and the Caribbean, or Africa). Over the last three years, the WGRC has initiated and led discussion on the research challenge of "data distillation" – referring to the challenge presented by the conflicting information from global climate models (e,g,, CMIP GCM runs), regional climate models (e.g., CORDEX runs), empirical-statistical downscaled data (e.g., statistical models using CMIP outputs as predictors), and multiple competing observational datasets of historical change and variability. It has also promoted a subtle yet important shift in emphasis from

"regional information" which puts the focus on data resolution for a location, to "information for regions" which recognizes that regions are related to climate processes at all scales.  The latter approach brings a holistic perspective to the climate drivers for regional decision-scale needs, and hence also for the VIA and climate service communities. The two themes of data distillation and information for regions are brought together in the concept of Frontiers of Climate Information (FOCI) projects which are designed to help advance the transformation of the multiplicity of data

products on climate change and variability into robust and scale-relevant information for decision needs.

### 4.1.4 CORDEX

The Co-ordinated Regional Downscaling Experiment (CORDEX; Giorgi et al., 2009) is a research project under the auspices of the WCRP with a vision to advance and coordinate the science and application of regional climate

downscaling through global partnerships.  CORDEX is principally focused on research using downscaling to better understand relevant regional/local climate phenomena as well as their variability and changes.  In the process

CORDEX seeks to improve regional climate downscaling models and techniques. Through regional teams CORDEX has been producing coordinated sets of regional downscaled projections for most regions of the world, and through the regional teams has fostered interaction with users of regional climate information. While there is high expectation

that CORDEX will provide more skillful projections for regions, the extent of added value from higher-resolution information is context-dependent and its use is complicated given limited resources within the VIACS and CORDEX communities to simultaneously explore multiple uncertainties including models, scenarios, and downscaling techniques. As such, the VIACS community should view CORDEX output as a valuable additional source of information that may be potentially incorporated alongside other data in the context of the WGRC's emphasis on

constructing "information for regions".

CORDEX has been successful in establishing regional research teams, and is currently in the process of establishing "Flagship Pilot Studies" (FPS) that will focus on targeted sub-continental regions to address key scientific questions and needs of the VIACS community. The current efforts are concentrated on developing phase 2 of CORDEX to

downscale from the CMIP GCMs to resolutions of 25km and higher using both dynamical and statistical downscaling. CORDEX is also developing ways to bring convergence between the RCM and empirical statistical downscaling (ESD) activities, and with GCM projections, in the context of the WGRC's distillation challenge.

### 4.2 Impacts Sector Communities

Research and applications communities have formed within a large number of impact sectors, offering an avenue of cohesive outreach for the VIACS Advisory Board. This section describes impact sectors' major focus, use of climate information, and community efforts for cohesive communication as an overview of the diverse VIACS communities and their unique needs for climate model outputs.

**4.2.1 Agriculture and Food Security**
Climate applications in the agricultural sector span sub-field-level support for management interventions to national and international level assessments of crop and livestock productivity, commodity prices, and food security. Climate information drives agricultural decisions on a continuum of time scales, with researchers and practitioners seeking to build systems that are sustainable and resilient to climate extremes, climate variability, and climate change. Climate

model outputs (particularly temperature, precipitation, humidity, and $CO_2$ concentrations) have long been used to drive agricultural assessments using a number of process-based and statistical approaches (Rosenzweig, 1984, 2014; White et al., 2011; Lobell and Burke, 2010; Asseng et al., 2013; von Lampe et al., 2014; Challinor et al., 2015). In recent years several groups have emerged to focus community efforts on agricultural impacts, including the Agricultural Model Intercomparison and Improvement Project (AgMIP, now encompassing 30+ activities;

Rosenzweig et al., 2015), and the Consultative Group on International Agricultural Research (CGIAR) Challenge Program on Climate Change, Agriculture, and Food Security (CCAFS; CGIAR, 2009). By connecting climate, crops, livestock, economics, and nutrition, the agricultural impacts community is engaged in many aspects of future scenario generation, integrated assessment, and decision support for a wide variety of actors (Rosenzweig et al., 2016). CMIP

outputs are a crucial element of most agricultural impact studies, which use a variety of downscaling and bias-
correction methodologies (White et al., 2011).

### 4.2.2 Fisheries and Marine Ecosystems

The ocean covers 70% of the Earth's surface, harbors rich diversity of species and ecosystems from the poles to the deep sea, provides 16% of animal protein consumed by humans globally, and supports the livelihoods for millions

(Mora et al. 2011, FAO 2014). Thus, the identification of climate change effects on marine ecosystems and the services they provide for human well-being is becoming increasingly important for management, conservation and food security (Merino et al. 2012, Barange et al., 2014). Over the past decades, various fisheries and marine ecosystem models have been created and applied to develop scenario-driven projections of future fisheries production (Blanchard et al. 2012), marine ecosystem structure and functioning (Jennings and Collingridge 2015) and species compositions

and distributions (Cheung et al. 2011). These individual models are often limited in scope (spatial, species, trophic group coverage), highly heterogeneous in terms of model structure, and dependent on the scientific or management question targeted. In addition, predicted outcomes are strongly dependent on which climate model is chosen to drive projections (Bopp et al. 2013), and so far there was limited choice among CMIP5 models due to missing data necessary to drive several marine ecosystem models. Also, GCMs are often poorly resolved in coastal oceans where most

fisheries production takes place (Barange et al. 2014). In 2013, the Fisheries and Marine Ecosystems Model Inter-comparison Project (FISH-MIP, https://www.isimip.org/gettingstarted/marine-ecosystems-fisheries/) was launched to systematically compare standardized climate scenarios across a broad range of both global and regional marine ecosystem models. During its development phase, FISH-MIP identified a number of missing variables now requested from CMIP6 via communication through VIACS (see Section 5.1 below) that would allow for greatly improved model

inter-comparison in the marine realm by including a wider range of GCMs and marine ecosystem models. FISH-MIP was also developed as part of the Inter-Sectoral Impact Model Inter-comparison Project (ISIMIP, see Section 4.3.1) to compare standardized climate scenarios across sectors, such as changes in food production on land and in the sea, terrestrial and marine biodiversity, and land-derived nutrient run-off affecting coastal ecosystems. Recently, two other marine model inter-comparison projects have been developed; the ICES/PICES Strategic Initiative on Climate Change

Effects of Marine Ecosystems (SICCME) and the Climate change and European aquatic RESources project (CERES). Both SICCME and CERES have a stronger focus on fisheries in selected regional ecosystems thus complementing the global focus of FISH-MIP. Together, these three initiatives – in conjunction with improved data availability from CMIP6 and communication via VIACS – will contribute to a better understanding of the impacts of climate change on fisheries production, marine biodiversity and ocean ecosystems.


### 4.2.3 Water Resources

Over the last couple of decades, there have been hundreds of studies into the impact of climate change on hydrological regimes and water resources (Jimenez Cisneros et al., 2014). The vast majority of these have been undertaken at the catchment or regional scale, using a wide range of hydrological models, water resources models and socio-economic

assumptions. These studies have shown that there is a wide diversity in estimated impacts of climate change, reflecting

variability in geographical context (in terms of hydrological regimes, management systems and demands on water resources), variability in the metrics defining impact, and variability in the methods and scenarios used to define future climate regimes. The construction of climate scenarios is central to hydrological impact assessments, and a wide range of techniques has been used to create scenarios at the appropriate spatial and temporal scales ("downscaling"). These

include the use of the delta method (applying projected changes to observed weather data), regional model output, bias-corrected regional or global model output, and stochastic weather generators. Whilst there have been attempts to inter-compare variants on a particular technique (e.g. different forms of bias correction), there have been no systematic assessments of the full range of potential methods at the catchment scale, or indeed of the full cascade of uncertainties on the magnitude and range of projected impacts. Comparisons between different studies in different locations are

made challenging by the use of different scenarios and downscaling techniques. There has historically been little coordination between groups in different locations assessing climate change impacts at the catchment and regional scale, although the UNESCO FRIEND-Water international collaborative hydrological programme (van Lanen et al., 2014) has a component seeking to undertake coordinated hydrological assessments of the effects of climate and other changes and several recent studies show the potential of model comparisons across scales (Hintermanns et al., 2016;

Gosling et al., 2016). There is greater coordination amongst the much smaller community of researchers assessing impacts on hydrological regimes and water resources across the global domain. The WaterMIP exercise inter-compared global hydrological model simulations using consistent data sets of current climate (Haddeland et al., 2011) and assessed the relative effects of hydrological and climate model uncertainty on changes in hydrological regime (Hagemann et al., 2013). More recently, ISIMIP (see Section 4.3.1) has involved an intercomparison of models and

projected changes using a wider range of hydrological models and climate scenarios (Schewe et al., 2014).

**4.2.4 Cities and Infrastructure**

The world's population is more than 50 percent urban and growing (Hunt and Watkiss, 2011; Rosenzweig et al. 2011), with many of the largest concentrations in coastal regions. High population density and growth can enhance

vulnerability and impacts. For example, in some cities rapid growth is concentrating more and more people in marginal areas, such as floodplains, while expansion of impervious surfaces further enhances flood risk. Other vulnerabilities include the health impacts of the urban heat island effect and poor air quality (Hunt and Watkiss, 2011). In many cities of the world, baseline information is lacking on both historical climate hazards (e.g., storm surge) and socio-economic information (e.g., population vulnerability), the latter in part due to rapid growth in those living uncounted in informal

settlements (Revi, 2008). Key climate information needs include observations and projections of 1) sea level change and coastal flood frequency and intensity, and 2) integrated measures of heat stress that go beyond temperature to consider joint hazards associated with humidity, and 3) other key extreme event metrics such as precipitation, drought, and wind intensity-frequency-duration (Blake et al. 2010). Due to large variations in micro-climate within cities (due for example to the urban heat island), high-resolution observational networks and remotely-sensed products are

needed. Downscaled projections such as outputs from regional climate models may be a valuable tool both 1) in regions where climate changes may be spatially heterogeneous (e.g. coastal regions) and 2) where there is a need for testing and evaluation of adaptation strategies at fine spatial scales (e.g. white-roofs or greening initiatives). As cities

have emerged as hubs for climate solutions, more organizations have been building networks and making urban-focused contributions. These include the International Council for Local Environmental Initiatives (ICLEI), the Urban Climate Change Research Network (UCCRN), and the C40 Cities Climate Leadership Group.

Diverse infrastructure types are also concentrated in and around cities as they are hubs of population and industry. Climate applications related to infrastructure are often challenged to identify the appropriate spatial resolution and domain given urban infrastructure corridors/networks and the large spatial signature of water- and infrastructure sheds that cities rely upon. For the energy sector, the relevant spatial scale may approach the continental. Much infrastructure is long-lived, capital-intensive, and geographically-fixed. These characteristics have encouraged the use of extreme event return periods in the design and financing of infrastructure. Key climate science questions are focused on how return periods for rare extremes such as the 1-in-100 year inland and coastal flood may change as the century progresses. Other climate hazards include extreme high temperatures, which for example can buckle, strain, and damage electrical and transportation systems as well as lead to weight restrictions in the aviation sector (Coffel and Horton, 2015). Minimum temperatures and related freeze-thaw cycle and icing issues also have large impacts on infrastructure. Many of the infrastructure-relevant climate needs are scientifically challenging due to their fine spatial scale and infrequency of occurrence, both of which amplify the signal of natural variability relative to climate change.

**4.2.5 Human health and well-being**

Weather and climate are among the drivers of a wide range of climate-sensitive health outcomes, including their incidence, geographic range, and seasonality (Smith et al. 2014). The sector is increasingly using climate information for risk management, particularly for developing early warning and response systems. Key weather and climate variables vary by health outcome, from relatively simple measures of daily temperature and precipitation for adverse health impacts from heatwaves and flooding, respectively, to more complex variables spanning seasonal to annual cycles, such as combinations of minimum and maximum weekly to monthly temperature with seasonal maximum and minimum precipitation to determine thresholds for outbreaks of malaria and other infectious diseases (e.g. Drake and Beier, 2014; Tonnang et al., 2010). There are few health outcomes for which there are multi-model projections of risk based on comparable assumptions, time slices, and scenarios (Caminade et al. 2014). Modeling the health risks of climate change is challenging because, in addition to weather and climate variables, multiple, interacting factors determine the overall health burden by affecting vulnerability, such as urbanization trends that affect urban heat islands, access to safe water, and other critical services; and by affecting the ability of communities and nations to prepare for and manage adverse health outcomes (Ebi and Rocklov, 2014). However, there are limited fine-scaled projections for many of these factors and their interactions. Different socioeconomic development pathways will lead to different levels of underlying vulnerability that will affect future health burdens (Ebi, 2013). Constructing scenarios with different combinations of emission and development pathways is needed to span the range of possible futures. Because many of the drivers of health outcomes arise in other sectors, efforts are needed to link health models with models of how climate variability and change could affect, for example, food- and water-security, energy production, land use, and ecosystem services.

### 4.2.6 Terrestrial Ecosystems

Climate impacts on ecosystems cover a range of biological and landscape features and management challenges ranging from biodiversity conservation, habitat changes, disturbance patterns, and ecosystem processes and services (such as carbon, nitrogen, and other biogeochemical fluxes and freshwater resources). A number of recent studies present evidence of climate change impacts on ecosystem aspects, and together they indicate increasing vulnerability across numerous taxa and ecosystems which are being affected.

Given this diversity of impacts on various ecosystem services, it is inherently important to develop climate services in collaboration with the community managing these ecosystem services at scales that their decision making and management units exist. As an example of a recent effort, in the US various agencies, including the Department of Interior (US DOI), US Department of Agriculture (USDA), and National Ocean and Atmospheric Administration (NOAA), a set of collaborative efforts is ongoing between the research community and the management community structured around regional centers enabling more focused dialogue for delivery of climate services. What has emerged from these interactions has been a more nuanced dialogue between the practitioners in the field and climate change applications researchers (e.g., McNeeley et al., 2016). This has enhanced understanding of constraints embedded in current climate projections and the temporal and spatial scale of ecosystem management decisions across various ecosystem services. Internationally, there are examples of efforts, such as those led by the GFCS and PROVIA which are providing information at scales to better understand ecosystem vulnerabilities to climate change, as well as to other critical sectors.

Ecosystem vulnerability studies and guidance to the management entities are challenged to provide climate information which are consistent across multiple scales in time and spatial extent. The climate information of seasonal characteristics and sensitivities related to variability of extreme events under differing climate realizations are useful to ecosystem level impact analyses. Efforts to develop these products with the user community is an ongoing process which the VIACS Advisory Board can further enable.

### 4.2.7 Other impacts sectors

Additional impact sectors are not strongly represented by current members of the VIACS Advisory Board despite considerable research and applications activity. These include the forestry and energy (e.g., wind and solar power generation as well as water resources for plant operations) sectors. The VIACS Advisory Board is eager to develop strong points of contact within these sectors to enhance communication with CMIP6 and other VIACS communities, and will look to bring in leaders from these sectors in the next Board term.

### 4.3 Integrative Communities

Communities that integrate physical and multi-sectoral research provide another resource that the Advisory Board utilizes to solicit VIACS expertise.

### 4.3.1 ISIMIP

Climate change will simultaneously impact different sectors. Projection of aggregated effects and an accounting for interactions, trade-offs, or co-benefits requires cross-sectorally consistent simulations (i.e., climate impacts projections that are forced by the same climate input data and based on the same story lines). The Inter-Sectoral Impact Model Intercomparison Project (ISIMIP; Warszawski et al., 2014; www.isimip.org) is designed to support the generation of these consistent projections through a common cross-sectoral protocol that could be integrated into the simulation protocols of sectoral initiatives such as the ones listed above. Analogously to CMIP, the simulation data are provided to all kinds of users in an open repository and the project is organized in different modelling rounds that will be dedicated to individual focus topics that will be selected by the impacts modelling communities and the users of the simulations (Rosenzweig et al., in review).

### 4.3.2 The Integrated Assessment Modeling Consortium

The Integrated Assessment Modeling Consortium (IAMC; http://www.iamconsortium.org) was created in 2007 in response to an IPCC call for a research organization to lead the integrated assessment modeling community in the development of new scenarios that could be employed by climate modelers for a new generation of climate change and related VIA projections. Its core missions include fostering the development of integrated assessment models (IAMs), peer interaction and vetting of research associated with IAMs, and the conduct of research employing IAMs, including model diagnosis, intercomparison, and coordinated studies. Most importantly, the IAMC promotes, facilitates and helps to coordinate interactions between IAM community and research communities studying climate change including climate modelers, VIA researchers, and technology and engineering communities. The IAMC has been active together with the International Committee On New Integrated Climate change assessment Scenarios (ICONICS) in establishing the overall conceptual framework and architecture for representative concentration pathways (RCPs; van Vuuren et al., 2011) and shared socioeconomic pathways (SSPS) (O'Neill et al, 2014; van Vuuren et al., 2014; Kriegler et al., 2014) and organized the development of the quantitative projections of the SSPs (Riahi et al., 2016), which will serve as inputs into CMIP6 climate and VIA assessments.

### 4.4 Climate Services Community

Many international, national and regional organizations exist to bring forward the development of climate services. The Roadmap for Climate Services of the European Commission (2015) defined 4 models of climate service providers: (1) Governmental cooperation/framework; (2) Extension of meteorological services; (3) Public climate services; and (4) University/groups of universities. We extend these here to recognize coordinated funding activities: (5) Private business development; and (6) Incorporation in business consultancy.

Various regional initiatives exist on climate services. The European Roadmap for Climate Services has a market-based approach, aiming to grow the demand for climate services, build a market framework (including standards) and also to enhance the availability and relevance of climate information (European Commission, 2015). The Copernicus

Climate Change Service (http://climate.copernicus.eu/) was also awarded in 2016 and tenders are currently under way to prepare the components including seasonal forecasts, climate data at global and regional levels, and economic and societal information for various impacts sectors. In the developing world the focus is more on improving availability of data to produce climate services products, reflecting recognised gaps (e.g., African Climate Policy Centre, 2013). In Africa, for example, the Climate for Development in Africa programme (under WMO Global Climate Observing System) and UNDP-led Programme on Climate Information for Resilient Development in Africa are playing a role in particular on the supply side of climate services. At the same time, there is increasing interest on the nature of demands for climate services.

At the first International Conference on Climate Services (ICCS) in 2011, participants agreed to form an open and informal coalition, the Climate Services Partnership (CSP), to improve the provision and development of climate services worldwide. The CSP has subsequently developed a paper on the ethics of climate services (CSP, 2015) and a review of on economic valuation of climate services (USAID, 2013). It continues its dialogues through annual ICCS (Vaughan, 2011; CSP, 2012; Lustig et al., 2014; Vaughan et al., 2015).

As a result of a decision made at the 2009 third World Climate Conference, in 2014 a Global Framework for Climate Services (GFCS; WMO, 2014) was established that is overseen by an Intergovernmental Board on Climate Services (IBCS). GFCS is supported by the CSP and operationally implemented by WMO with the aim of "providing climate information in a way that assists decision-making by individuals and organizations". GFCS has identified five priority sectors – agriculture and food security, disaster risk reduction, energy, health, and water – and is supporting projects in these areas around the world with a focus on developing services through engagement with users. A goal for the VIACS Advisory Board is to establish a formal relationship with the GFCS to better communicate between the climate services and climate modelling communities.

**5 VIACS Activities**

Since its launch in 2015, the VIACS Advisory Board has engaged the CMIP community on several issues summarized here to illustrate the types of interactions and information that this new conduit of communication enables.

**5.1 Prioritization of CMIP experiments and outputs**

On request from the CMIP6 leadership, the VIACS Advisory Board tasked its members to solicit feedback from their respective communities as to the variables and experiments of highest priority for their planned applications of CMIP6 model output. This feedback benefits the CMIP modeling groups in that they can determine the potential for variables or experiments to be used by different applications groups. In response, the VIACS Advisory Board constructed a single spreadsheet with the set of more than 900 CMIP5 variables and the list of 188 proposed CMIP6 MIP experiments and requested that VIACS experts prioritize sets of variables and the experiments they are interested in exploring via a template. This spreadsheet was distributed through the Board members to many VIACS communities along with a document detailing the request for input in the CMIP6 planning process. It is clear that the large number

of variables and experiments was daunting to some VIACS experts, so the VIACS Advisory Board received a mixture of spreadsheet and more generally-written feedback. Key messages emerged in the VIACS community response:

*Key Message 1: Core variables were already in CMIP5 for most VIACS needs. Some communities requested*
*different sets of variables, additional skill metrics, and increased validation of GCM outputs against observations.*
Many of the VIACS groups reported the key variables for impacts assessment were already present in CMIP5 and wished to see them continued in CMIP6. Chief among these were temperature, precipitation, radiation, and humidity variables at daily and monthly time scales, which were requested by nearly all communities. Beyond these core variables there is a tremendous diversity in variables requested across impacts sectors, although the
680 majority of these variables were already in the CMIP5 variable list. It was not practical to merge these variable lists into a single priority list, as variables that are of high priority for one impacts sector may not be needed by another. Groups also indicated that modeling groups should consider variable sets in addition to isolated variables, as some applications need a complete set of variables to proceed (e.g., mitigation studies need a set of variables related to land use and carbon content but are challenged to proceed if some are missing; statistical methods may
only be possible if a set of variables are available). Many of the groups requested that the climate modeling community enhance analysis of these variables' biases (e.g., biases in projected regional changes of humidity or solar radiation) and develop guidance for VIACS applications that must deal with these biases.

*Key Message 2: New variables are needed by some VIACS communities.*
The agricultural, fisheries, energy, and climate services communities requested additional variables, as detailed in Table 3. These include entirely new variables, altered temporal resolution for existing variables, and capture of sub-grid-scale information that is otherwise lost in aggregation. To better understand extreme events and their impact on agriculture, energy, urban areas, health, and climate services in many sectors, statistics of high-frequency events could be provided at a monthly scale. Examples include the average precipitation rate on days
where precipitation occurred paired with number of precipitation days, the maximum 2-hourly precipitation total in a given month, or wind gusts at various altitudes (for wind power applications). These additional variables were most often ranked in the highest priority set and requested for the Historical, DECK, and ScenarioMIP experiments, although requests include experiments from 12 of the 17 CMIP6-Endorsed MIPs. Although the VIACS Advisory Board does not itself perform any model output analyses, groups responding to the VIACS
Advisory Board request indicated a commitment to analyze requested outputs.

*Key Message 3: Several groups indicated that high-resolution variables may be best produced through downscaling rather than directly from global climate models, but that it would also be helpful to have the GCM outputs as a basis for comparison.*
Several groups detailed the variables needed to run their impacts models, but also indicated that they expect to draw their inputs from statistical scenarios or from CORDEX (or other regional climate model) results (often with additional bias correction) rather than from the global models themselves. This is particularly true for temperature

and precipitation extremes as well as water and energy balance variables related to hydrology, agriculture, energy, and coastal processes.  In a similar manner, climate service providers (in particular) noted that the monthly outputs provided by CMIP in previous IPCC Assessment Report phases were not as desirable; daily (or sub-daily) time scale is of greatest interest.  This opinion is not universally held, but more variables at daily resolution would be welcomed, with overall archive size depending on the level of interest and utility within the VIACS community.

*Key Message 4: The experiments of greatest interest are the Historical Simulation, the DECK experiments, the RCPs within ScenarioMIP, and the hindcasts and forecasts of the Decadal Climate Prediction Project.*
Members of the VIACS Advisory Board also expressed an interest in providing societal implications for CMIP6-Endorsed MIPs, for example including the development of RCPs (van Vuuren et al., 2011) and SSPs (O'Neill et al., 2014; Riahi et al., 2016) with ScenarioMIP, the use of ecosystem and agricultural models in conjunction with LUMIP, the health impacts of pollution policies in AerChemMIP, or the role of water resource management in LandMIP.  In many cases the CMIP6-Endorsed MIPs contain experiments that explore specific physical relationships within the climate system, and only a subset is directly relevant to societal applications. VIACS researchers and practitioners often expressed interest in this small subset of experiments (or even one single experiment) from a given MIP's experiment group, which will help modeling groups determine an efficient provision of the requested outputs while avoiding comprehensive variable lists where there is little interest in a large portion of the data.  Only the Radiative Forcing MIP did not have any experiments specifically requested for sectoral application in the VIACS solicitation.

As a result of the VIACS Advisory Board's request, the CMIP6 data archive may now be searched according to variable packages indicated with different priority levels for each responding VIACS community.  For example, seven different packages exist for the AgMIP community, including a package containing the necessary variables to drive crop models and a package that would facilitate the closing of carbon budgets in agricultural areas.

**5.2 Obs4MIPs**
CMIP6 leadership requested input from the VIACS community about observational datasets utilized by various VIACS sectors that could be used as additional sources of validation for climate model output as part of Observations for Model Intercomparisons (Obs4MIPs). The WCRP's Data Advisory Council (WDAC) Observations for Model Evaluation Task Team curates these Obs4MIPs datasets to improve model evaluation and process understanding.

The VIACS Advisory Board found that, in general, there were only a few recommendations for new data sets to include in Obs4MIPs.  One concrete example was to better compare climate output with observations related to snow for a variety of applications including water resources. There are a number of satellite-based data products such as those from the Globsnow project (providing northern hemisphere daily snow extent and snow water equivalent; Metsämäki et al., 2015) that have not yet intensively been compared to climate model output.  It would be useful to look at crop season and yield databases (e.g. Ramankutty et al., 2008; Monfreda et al., 2008; Ray et al., 2015) to better

align seasonal variation in productivity, greenness, and soil moisture over agricultural lands against climate models' vegetation/land-surface model outputs (which often represent crops as generic grasses that lack the observed sequences of crop and fallow periods).

The VIACS Advisory Board also discussed the potential creation of an equivalent to Obs4MIPs for the VIACS
communities, facilitating validation and process understanding for sector models.  For example, this could include recently-created datasets for agriculture such as time series of yield (Ray et al., 2015), fluorescence (Joiner et al., 2014), and above-ground biomass (Tucker et al., 2005).  European climate services also indicated an interest in more closely aligning efforts to compare with the Copernicus operational satellite services being developed by the European Commission.  Many VIACS communities have opportunities to coordinate efforts on climate-related datasets even if
they are not directly comparable to climate model outputs.  This new "Obs4VIACS" could potentially be an element of Obs4MIPs or could be organized as a parallel effort.

**5.3 Gridding of GCM outputs**

The VIACS Advisory Board also solicited feedback on a CMIP6 data request seeking input on the extent of
harmonization that was needed for model output grids.  At issue was the contrast between raw climate model output (which may come on irregular and/or unique grids) and the need for a regular and harmonized grid for applications purposes.

Feedback indicated that the VIACS communities are interested in GCM outputs eventually reaching a common grid
for model intercomparison and multi-model applications, and that regular grids are most useful for these purposes. This is particularly true because VIACS communities often utilize multiple climate output variables and observational data sets.  It is therefore desirable to have a smaller number of necessary conversions, and useful to have common methods for multiple variables. Many groups have developed techniques to re-grid and/or interpolate to common grids (often ~0.5x0.5 degrees), but several groups indicated that it would be preferable to have CMIP or other climate
experts perform this re-gridding so that it could be quality-controlled and consistent across applications. This work could begin with those output variables most commonly requested by VIACS groups (monthly temperature, precipitation, radiation, and humidity, adding wind speed would also enable Penman-Monteith potential evapotranspiration calculations).  Some common gridding and scenario generation was done within ISIMIP (Warszawski et al. 2014), but a central and community-driven effort would be welcome, particularly with regards to
extreme events that are vital to many sector analyses but are not captured well by some methods (e.g., Guentchev et al., 2016).

Although there was interest in the common grids, VIACS Advisory Board members also indicated an interest in the raw model outputs as these are needed to understand the physical basis and relationships among variables contained
in the outputs. Only providing harmonized and re-gridded outputs would limit the opportunity to test out the benefits of different methods for re-gridding that may be advantageous for different applications.  The VIACS Advisory Board

therefore requested that model outputs be provided in their native format and that CMIP initiate a re-gridding effort oriented toward producing a common and regular grid to facilitate applications.

**5.4 Future Activities**

Future activities of the Board will also support the creation of products that facilitate the use and uptake of climate model outputs for societal applications. VIACS guidance will support the development of online metrics and visualizations for the VIACS community of researchers, practitioners, stakeholders, and decision-makers (potentially thought platforms such as ESMValTool, Eyring et al., 2016b). These include metrics and derived variables made through a combination of climate outputs or sector-specific thresholds (e.g., frost-free days for agriculture, over-winter minimum temperatures for health and ecosystems, days of airplane weight restriction due to temperatures), potentially in collaboration with the Expert Team on Climate Change Detection and Indices (Sillmann et al., 2013a, 2013b). Although the production of guidance documents is beyond the purview of its mandate, the VIACS Advisory Board will help determine requirements for documentation and online guidance that will facilitate the use of CMIP6 products by various user communities. This could include contributing to formal surveys of the VIACS and climate modeling communities in order to identify cross-cutting engagement needs (within CMIP, PROVIA, or the Climate Services Partnership, for example). The Board will also encourage the inclusion of both climate modeling and climate applications experts in the generation of vetted, bias-corrected, accessible, and appropriately-formatted climate model outputs for use in VIACS research and for distribution on climate information portals created by knowledge providers. In addition, it will promote further evaluation and transfer of good practices in CMIP output application within the VIACS community, including the assessment of uncertainty propagation as information cascades from climate to VIACS models and assessments and its potential feedback effects on the climate system. The Board is well-positioned to provide VIACS facilitation on climate model simulations and analyses for future IPCC assessments and special reports, including the upcoming 1.5 $^{\circ}$C assessment, and encourages engagement around broader discussions about the extent to which i) more and improved climate model outputs add value both to impact models, and ii) more and improved climate and impact model outputs add value to impact sector decision making (Dessai et al. 2009).

**6 Summary and Benefits**

The VIACS Advisory Board was created as an element of CMIP6 to facilitate communications between the climate modeling community and the scientific and operational communities that apply climate model output for societal benefit. Launched in 2015, the VIACS Advisory Board developed a framework to interact with the CMIP6 leadership, convene experts of the VIACS impact sectors and programs, and solicit wider input from the broader communities they represent. The VIACS Advisory Board facilitates efforts to address all three key science questions of CMIP6 because the VIACS community has an acute interest in the best possible information about (1) how the Earth System (in particular the impacted elements relevant to society) responds to forcing, (2) how model biases potentially influence decision-making in impacted sectors, and (3) how climate variability, predictability, and uncertainty may be handled in preparing climate change adaptation and mitigation strategies that benefit impacted sectors. Initial activities demonstrate the utility of this approach in the identification and prioritization of CMIP6 output variables

and MIP experiments for VIACS applications, and Board inputs are also expected as visualization and communication
products are created to further disseminate CMIP6 outputs to the applications community.  Interaction related to the
design and prioritization of model output variables has already led to tangible progress including the creation of model
output packages tailored according to the requests of VIACS communities that participated in the initial request for
input.

The VIACS Advisory Board will be most successful if it is utilized by both the climate modeling and climate
applications communities.  Cognizant of continuing (and in many cases healthy) differences in interests, priorities,
and expertise between the climate modeling and applied climate communities, the VIACS Advisory Board aims to
highlight opportunities for coordination that facilitates collaboration and overall benefit to both science and society.
A continuing challenge will be the identification of contact points and networks that allow for broad and inclusive
interaction, as well as maintaining willingness within the communities to respond to requests in a timely manner.  The
VIACS Advisory Board alone cannot overcome all gaps, however the Board is designed to benefit a number of
communities that engage in CMIP6 and applications efforts, and aims to synthesize contributions beyond the sum of
its individual interactions.

*Potential benefit to the climate modeling community.* The VIACS Advisory Board has already provided advice on
important climate variables to be requested from climate modelers, including downscaled information, for use in
VIACS analyses. The Board aims to improve the relevance of climate model outputs to society through the
development of more creative, robust, and efficient applications of climate model outputs. The Board also facilitates
dissemination of important scientific findings and model-specific caveats that need to be recognized in the design and
communication of climate impact assessments.

*Potential benefit to the Vulnerability, Impacts, and Adaptation (VIA) and Climate Services (CS) communities.* The
VIACS Advisory Board seeks to enhance substantially the level of communication between CMIP and the VIACS
community, with mutual benefits. In particular, the Board communicates and disseminates information to the VIACS
community regarding access to, and understanding of, key climate model and related scenario outputs for VIACS
research and wider societal applications.  The Board also helps improve linkages across the IPCC Working Groups.

*Potential benefit to the Integrated Assessment Modelling (IAM) community.* Beyond their role in exploring mitigation,
IAMs also represent climate change impacts and adaptation, albeit in simplified form. The IAM community relies on
results and insights from VIACS studies to test and calibrate their models. Moreover, IAMs can provide valuable
information to VIACS applications that also require scenarios of socioeconomic and/or land use change concurrently
with climate projections. The VIACS Advisory Board has the potential to advise on important socioeconomic
variables to be requested from global IAMs that are consistent with climate projections generated in the CMIP6
process, most notably through interactions with SceanrioMIP (O'Neill et al., this issue).


*Potential benefit to policymakers.* The VIACS Advisory Board has the potential to help CMIP6 incorporate the experience of VIACS community interactions with policy-makers around the world, with plans for online metrics tailored toward policymakers and a greater translation of climate model output toward societally-relevant outcomes that are central to policymaker interests.


**Data Availability:** As a diagnostic and advisory contributor to CMIP6, the VIACS Advisory Board does not generate new data or model output. Variable packages for each VIACS community that responded to the variable request may now be specifically requested at http://clipc-services.ceda.ac.uk/dreq/u/VIACSAB.html. Documentation of community engagement and feedback is provided to CMIP6 leaders, and is available upon request. The VIACS

Advisory Board is also developing a website to house information about the Board and documentation of communications activities, which will be linked to the main CMIP webpage (http://www.wcrp-climate.org/wgcm-cmip/wgcm-cmip6).

**Acknowledgements:** The authors are grateful for contributions of the VIACS communities, in particular those who
responded to requests for information related to variables and experimental simulation priorities, observational datasets, and gridding needs. We also thank Martin Juckes for his engagement regarding the development of VIACS variable packages within the CMIP archive structure. We also acknowledge participants at the Aspen Global Change Institute on the Experimental Design of CMIP6 in August, 2014, which contributed initial conversations on the potential of an Advisory Board. Dr. Ruane's work was supported in part by the NASA Modeling, Analysis, and
Prediction Program.

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

**Table 1. Summary of the CMIP6 DECK and CMIP6-Endorsed Model Intercomparison Projects (MIPs). More detail about CMIP6 organization is provided by Eyring et al. (2016), and each of these CMIP6-Endorsed MIPs is described in more detail in a separate contribution to this Special Issue.**

| *Short Name* | *Long name* | *VIACS community expressing interest in at least one experiment [a]* |
|---|---|---|
| *Central Set* | | |
| **Historical** | CMIP6 Historical Simulation | All |
| **DECK** | Diagnostic, Evaluation and Characterization of Klima | All |
| *CMIP6-Endorsed MIPs (each contains a set of experiments)* | | |
| **AerChemMIP** | Aerosols and Chemistry Model Intercomparison Project | Agriculture, Terrestrial Ecosystems, Health |
| **C⁴MIP** | Coupled Climate Carbon Cycle Model Intercomparison Project | Ag, Fisheries, Marine Ecosystems |
| **CFMIP** | Cloud Feedback Model Intercomparison Project | Fisheries, Marine Ecosystems |
| **DAMIP** | Detection and Attribution Model Intercomparison Project | Agriculture, Fisheries, Marine Ecosystems, Climate Services |
| **DCPP** | Decadal Climate Prediction Project | All |
| **FAFMIP** | Flux-Anomaly-Forced Model Intercomparison Project | Fisheries, Marine Ecosystems |
| **GeoMIP** | Geoengineering Model Intercomparison Project | Agriculture, Fisheries, Marine Ecosystems |
| **GMMIP** | Global Monsoons Model Intercomparison Project | Fisheries, Marine Ecosystems, Terrestrial Ecosystems |
| **HighResMIP** | High-Resolution Model Intercomparison Project | Fisheries, Marine Ecosystems |
| **ISMIP6** | Ice Sheet Model Intercomparison Project for CMIP6 | Fisheries, Marine Ecosystems |
| **LS3MIP** | Land Surface, Snow and Soil Moisture | Terrestrial Ecosystems |
| **LUMIP** | Land-Use Model Intercomparison Project | Agriculture, Terrestrial Ecosystems, Climate Services |
| **OMIP** | Ocean Model Intercomparison Project | Fisheries, Marine Ecosystems |
| **PMIP** | Paleoclimate Modelling Intercomparison Project | Fisheries, Marine Ecosystems |
| **RFMIP** | Radiative Forcing Model Intercomparison Project | None |
| **ScenarioMIP** | Scenario Model Intercomparison Project | All |
| **VolMIP** | Volcanic Forcings Model Intercomparison Project | Agriculture |
| *CMIP6-Endorsed Diagnostic MIPs (no experiments, but specific analyses planned)* | | |
| **CORDEX** | Coordinated Regional Climate Downscaling Experiment | N/A |
| **DynVarMIP** | Dynamics and Variability Model Intrecomparison Project | N/A |
| **SIMIP** | Sea Ice Model Intercomparison Project | N/A |
| **VIACS AB** | Vulnerability, Impacts, Adaptation and Climate Services Advisory Board | N/A |

[a] Not all VIACS communities weighed in on initial variable and experiment request; dialogue ongoing.

**Table 2. Five steps followed for each VIACS Advisory Board consultation to focus on CMIP/VIACS communications. If the VIACS community requests information from the CMIP community, a similar process would is conducted in the opposite direction.**

| Step | Description |
|---|---|
| 1 | VIACS Advisory Board Co-Chairs reach out to CMIP6 representatives to solicit input, requests, or questions to propose to the VIACS Advisory Board (via email or teleconference). |
| 2 | VIACS Advisory Board Co-Chairs prepare summary documents or worksheets that provide a coherent template for the solicitation of input across the VIACS communities. |
| 3 | The VIACS Advisory Board holds a teleconference to discuss the CMIP6 questions, request solicitation of information using the provided templates, and raise issues from the VIACS communities. |
| 4 | Board members survey their respective networks of colleagues and provide collated responses back to the Co-Chairs. |
| 5 | Co-Chairs submit a summary of the CMIP6/VIACS community interactions, solicitation results, and action items identified by the Board to all Board Members and the CMIP6 leadership (to be shared with MIP leaders as needed). |


**Table 3: Additional variables requested through the VIACS Advisory Board process. Note that the solicitation allowed each respondent to nominate variables of interest, but additional work is needed to iterate and gauge interest on these variables across all of the VIACS communities.**


| Time Resolution | Name (plus description as needed) | Units | Additional Notes |
|---|---|---|---|
| *New variables requested by Agricultural sector (for Historical, DECK, and ScenarioMIP experiments, as well as requests for experiments within AerChemMIP, C⁴MIP, DAMIP, DCPP, GeoMIP, LUMIP, and VolMIP).* | | | |
| Monthly | surface concentration of Ozone | kg m-3 | Also for use ecosystem and health sectors |
| Daily, monthly | cropland tile maximum temperatures | K | |
| Daily, monthly | cropland tile minimum temperatures | K | |
| Daily, monthly | cropland tile precipitation | K | Tile contains information from agricultural fraction of land in a given GCM grid box. |
| Daily, monthly | cropland tile minimum relative humidity | K | |
| Daily, monthly | cropland tile wind speed | K | |
| Monthly | number of precipitation days where accumulation was above 1 kg m-2 | # | These two variables combine to describe the intensity of rainfall when it does occur |
| Monthly | average precipitation accumulation on days where accumulation was above 1 kg m-2 | kg m-2 | |
| *New variables requested by Fisheries and Marine Ecosystems sectors (for Historical, DECK, C⁴MIP, DAMIP, FAFMIP, GeoMIP, OMIP, and ScenarioMIP experiments, as well as requests for experiments within DCPP and ISMIP).* | | | |
| Monthly | Photosynthetic active radiation (PAR, 400-700nm) | W m-2 | |
| Monthly | Euphotic depth 1 = depth at which there is 1% of surface PAR | M | |
| Monthly | Euphotic depth 2 = depth at which the PAR is 0.1 W/m2 | M | |
| Monthly | 3-D (depth-resolved) ocean temperature | K | |
| Monthly | 3-D (depth-resolved) salinity | Psu | |
| Monthly | 3-D (depth-resolved) current velocity u | m s-1 | |
| Monthly | 3-D (depth-resolved) current velocity v | m s-1 | |
| Monthly | 3-D (depth-resolved) dissolved oxygen concentration | mmol m-3 | |
| Monthly | 3-D (depth-resolved) pH | pH | |
| Monthly | 3D (depth-resolved) primary productivity | mol C m-3 s-1 | |
| Monthly | 3D (depth-resolved) phytoplankton carbon concentration | mol m-3 | |
| Monthly | 3D (depth-resolved) small phytoplankton carbon concentration | mol m-3 | |
| Monthly | 3D (depth-resolved) large phytoplankton carbon concentration | mol m-3 | |
| Monthly | 3D (depth-resolved) zooplankton carbon concentration | mol m-3 | |
| Monthly | 3D (depth-resolved) small (micro-)zooplankton carbon concentration | mol m-3 | |
| Monthly | 3D (depth-resolved) large (meso-)zooplankton carbon concentration | mol m-3 | |
| Monthly | 3D (depth-resolved) small particulate carbon concentration | mol m-3 | |
| Monthly | 3D (depth-resolved) large particulate carbon concentration | mol m-3 | |
| Model-specific | size ranges or Min-Max of phyto and zooplankton groups (would need to know the range of sizes for the biogeochem model variables; e.g. ESM2M has small and large groups) | mass ranges | |
| *New variables requested by Climate Services (for Historical and DECK as well as experiments within ScenarioMIP).* | | | |

| | | | |
|---|---|---|---|
| Not specified | Sunshine Duration | s | Defined using threshold value to determine intense sunshine |
| Not specified | Potential Evaporation | mm | ideally separately by land use (as calculated) |
| Not specified | Evapotranspiration | mm | |
| Not specified | $CO_2$ Concentration in near-surface layer | kg m-3 | Agriculture and ecosystems |
| Not specified | Wind Speed | m s-1 | Stored at model level not pressure level |
| Not specified | Wind Direction | Degrees | Renewable energy (wind) |
| Not specified | 100m Wind Speed and Gusts | m s-1 | Also 80m and 120m for energy resources and infrastructure |
| Not specified | 10m Wind Gusts | m s-1 | |
| | Wave Height Max | m | |
| 3- or 6-hourly | Geopotential Height | m | On more pressure levels 300, 500, 850, 925, 1000hPa |
| 3- or 6- hourly | Boundary Layer Height | m | |
| 3- or 6- hourly | Vertical Velocity | Pa s-1 | At more frequent output times |
| 3- or 6- hourly | Convective Precipitation | kg m-2 s-1 | solid and liquid separated |
| 3- or 6- hourly | Total Soil Moisture Content | kg m-2 | Possibly more layers |
| 3- or 6- hourly | Soil Temperature | K | At more frequent output times |
| 3- or 6- hourly | Relative Vorticity | s-1 | |
| 3- or 6- hourly | Relative Humidity | % | |
| 3-hourly | Mean Sea Level Pressure | hPa | At more frequent output times |
| 3- or 6- hourly | Large Scale Precipitation | kg m-2 s-1 | |
| 3- or 6- hourly | Eastward Wind | m s-1 | On more pressure levels |
| 3- or 6- hourly | Northward Wind | m s-1 | 300, 500, 850, 925, 1000hPa |
| 3- or 6- hourly | Specific Humidity | 1 | 300, 500, 850, 925, 1000hPa |
| 3- or 6- hourly | Snow Depth | M | At more frequent output times |
| 3- or 6- hourly | Snow Density | kg m-3 | Comment from Swedish Meteorological and Hydrological Institute: "everything related to snow is desired" |
| 3- or 6- hourly | Snow water equivalent | kg m-2 | |
| 1-hourly | Precipitation | kg m-2 s-1 | High frequency precipitation data |
| 3-hourly | Precipitable water in the atmospheric column | kg m-2 s-1 | |
| Monthly | Maximum accumulated precipitation over 1 hour | kg m-2 | Similarly, maximum accumulated precipitation over 1-, 2-, 6-, 12-, and 24-hour periods |
| Monthly | Maximum ocean wave energy | Not provided | |
| Monthly | Total atmospheric heat content | Not provided | |
| Monthly | Total oceanic heat content | Not provided | |
| Monthly | Total land heat content | Not provided | |
| Monthly | Total glacier heat content | Not provided | |
| *New variables requested by Energy sector (for Historical, DECK, and ScenarioMIP experiments, as well as requests for experiments within HighResMIP).* | | | |
| Daily Mean | 100m Wind Speed | m s-1 | Focus on wind speeds at 100m above surface |
| Daily Mean | Eastward 100m Wind | m s-1 | |
| Daily Mean | Northward 100m Wind | m s-1 | |

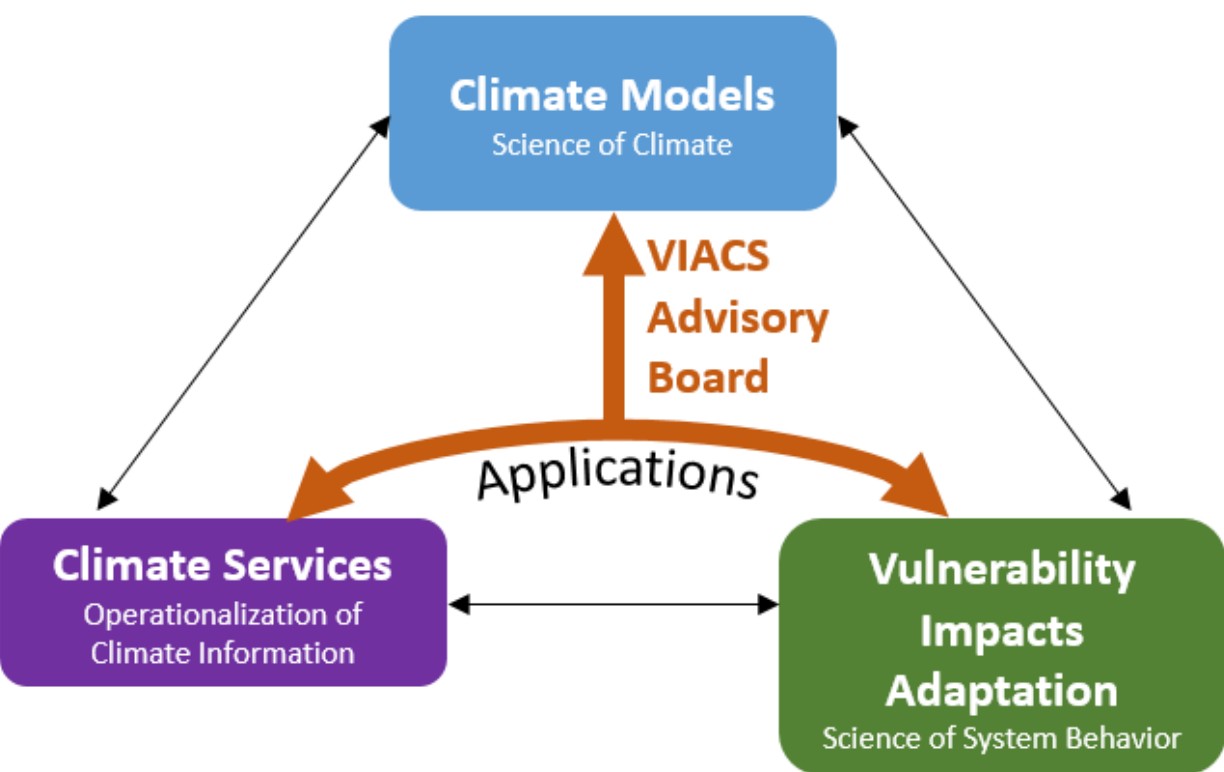

**Figure 1: The VIACS Advisory Board provides a new mechanism to help integrate the Vulnerability, Impacts, and Adaptation communities with the Climate Services community, allowing for more comprehensive communication between the climate modeling community and those who apply climate model outputs. Black lines represent previous lines of communication, with the VIACS Advisory Board now helping to connect applications communities and provide a conduit for communications with the climate modeling community.**



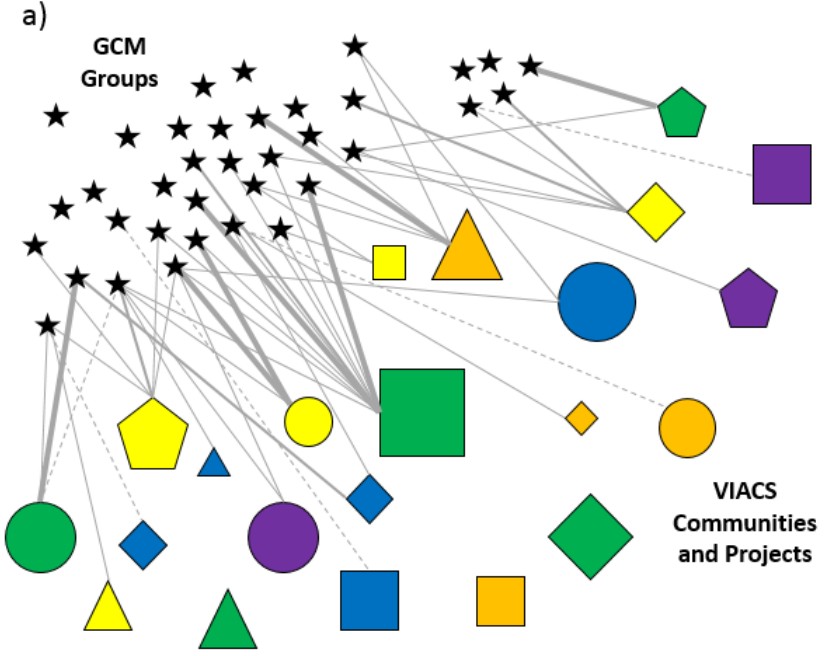

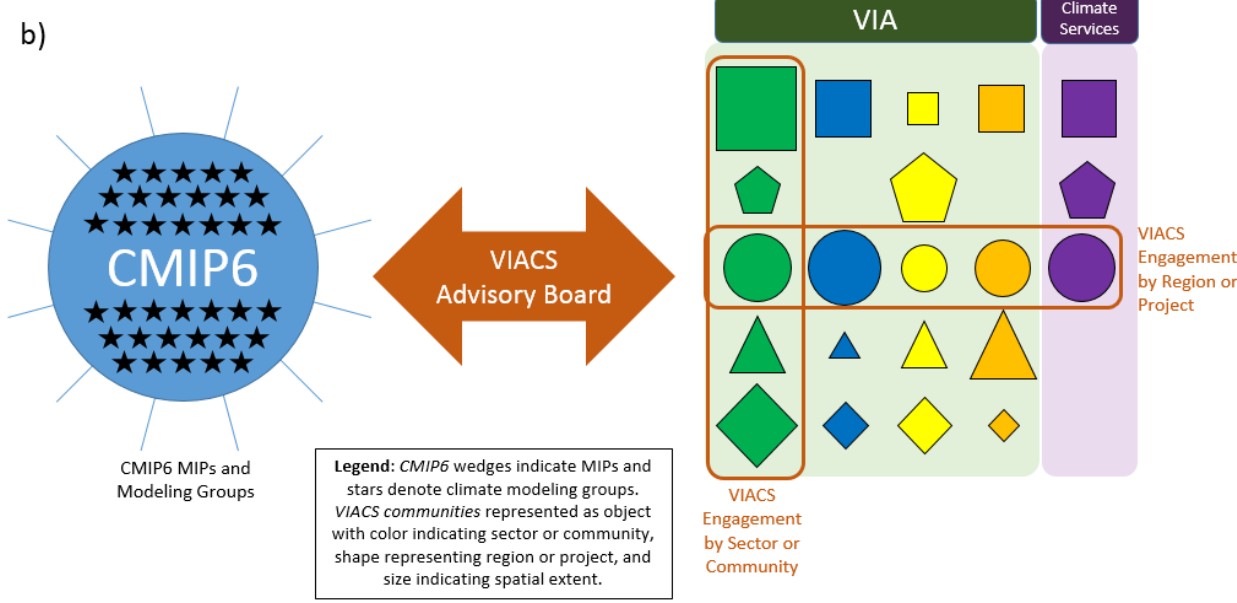

**Figure 2: Schematic illustrating the development of the VIACS Advisory Board as an organized process of communication between the climate modeling community and the climate application communities. a) Absent organized communication, each climate modeling center and each climate applications entity had to connect and maintain communications, resulting in a mixture of strong, convoluted, or absent lines of communication. b) As the climate modeling community has organized interactions through CMIP6 (and the CMIP6-Endorsed MIPs; Eyring et al., 2016a), the applications communities of VIA research and the emerging climate services community can utilize the VIACS Advisory Board to provide coherent interaction with CMIP6 leadership and modeling groups. Note that lines of communication are not equivalent to modes of data access, which would include various data distribution centers and clearinghouses.**

