# Peer review of "The Vulnerability, Impacts, Adaptation and Climate Services Advisory Board (VIACS AB v1.0) contribution to CMIP6"

_Geoscientific Model Development, 2016_

## Referee Comment (RC1) · Anonymous Referee #1 · 5 Apr 2016

The paper looks great and it covers various issues on the application of of CMIP6 coupled models for studies on VIACS. However, CMIP6 models do not include impact models, and without that it would be hard to perform VIA applications. I am happy that the PROVIA be part of this, but as far as I can see vulnerable regions such as SE Asia, Central and South America and Africa (countries other than South Africa) are out of the Advisory Board, as is shown in the paper. May by PROVIA can cover this gap but PROVIA is not the advisory board. The authors wrote that " CMIP6 provided a unique opportunity 35 to facilitate a two-way dialogue between CMIP6 climate modelers and VIACS". however, I can not see how this dialogue would come up in the paper.

In my experience, climate modelers work not so much on societal relevant issues, and the simulation outputs data not always are ready to use for impacts studies. the VIA community sometimes is not familiar with model ouputs. So, from what I see in the

paper, the probelms may continue, no matter how complex would be CMIP6 models.

Around line 290, "Representatives of the VIACS Advisory Board also participate in major CMIP6 meetings to give voice to the VIACS perspective", what kind of participation?, what kind of voice?. it is the voice of the members of of the VIA community that may not be part of this board?. How the VIACS board works with modellers?

Working on VIA myself, I realized that not everything is solved by climate coupled models, as those from CMIP5 or CMIP6, nor by regional climate models. Regional climate models and experiments such as CORDEX and others represent model applications that may generate data that would feed impacts models. The paper describes some sectors in which the VIA focus would be, but the authors would not say how this will be done. There should be a section on impact models and how uncertainties would be assessed. It is my suggestion that this paper's focus should be wider than the advisory committee, and it should reflect a global reality, by adding authors from regions such as India, China, Central or Southern Africa, Central and South America, so the global flavour would be on it. It would be nice that the advisory committee on VIACS be more regionally representative, but at least that paper must reflect the different realities, and I suggest including authors from vulnerable regions.

---

## Referee Comment (RC2) · Anonymous Referee #2 · 12 Apr 2016

This is paper is useful, informative, and well written. I appreciate the work of the authors in putting it together.

The paper is particularly successful at providing an overview of important communities engaged in work at the nexus between climate and VIA research, and in shedding light on recent activities of the VIACS AB in facilitating communication between these different communities.

A few thoughts on how this draft might be improved:

***1) I'm not sure that either the abstract or the introduction provides an accurate map of the paper?

I would have expected some early part of this paper to say something like "This paper

describes the motivation that led to the development of the VIACS AB, provides an overview of the various communities it attempts to engages, and summarizes recent activities." Or something similar. For this reader, it wasn't entirely clear where the paper was going until the end.

***2) I'm also curious if this paper could be a bit more ambitious in offering a vision for the VIACS AB?

I see this paper as saying that the VIACS AB will facilitate communication between disparate communities, and then summarizing some recent activities to that end.

This is fine, but I'm wondering if the paper couldn't go a bit further in synthesizing what sorts of information / messages / lessons the VIACS AB has learned from different kinds of communities? And can the paper identify some major questions that need to be resolved or addressed by VIA researchers engaging climate modelers?

Pulling this out of section 4, where the state of work in various communities is described, and out of section 5 (particularly key messages from the prioritization activity) would provide a sense of key issues that this group will need to tackle and a greater perspective on the orientation of the co-chairs. It would also offer a more compelling conclusion, offering a bridge between the summary section and the benefits.

Though it's coming a bit off the cuff, I'm also wondering if there's some way to link that kind of synthesis to the three science questions of CMIP, or their VIA interpretation on p 7?

***3) On a related topic, I'm wondering if there's scope to propose future activities for the AB?

I see that the paper suggests establishing a formal link with the GFCS, and that the conclusion section indicates that the VIACS AB will be most successful if it identifies contact points and networks that allow for a broad and inclusive interaction. It may also be that section 5.4 is describing future, rather than present / past, actions.

But I'm wondering if there's something more that can be said? Are the authors able to articulate some priority actions that would give readers a clearer sense of what they see as most important steps? In many cases, this may just be a matter of distilling material that appears earlier, a bit less directly, into the conclusion section.

From my perspective, this kind of distillation would provide readers with a more concrete sense of what the board plans to do, and an easier read.

***4) Will there be a follow up paper that addresses how / whether CMIP addressed the guidance it got from the AB?

The key messages section is really interesting . . . I'd be interested as well to hear what actions were taken in response to the advice provided. Is any of that available now?

Addressing a few of these issues would force the authors to synthesize things a bit more, and to offer perhaps a more elaborated view of the role they see the VIACS AB playing in the future. I think this would add value to the paper and provide the reader a better sense of the board members' vision.

---

## Referee Comment (RC3) · Anonymous Referee #3 · 8 May 2016

General comments

The VIACS Advisory Board is an excellent initiative. It addresses a key gap between climate modellers and the user community, particularly in the context of the CMIP ensembles. This paper describes the VIACS AB, including how it is constituted, its scope, mode of operation, objectives etc. The paper does this well and is a worthy publication, requiring little in the way of significant modification. Following here are some specific comments for the authors and readers to consider. I note that some of these may apply more to the VIACS AB operation in general than to this manuscript in particular. Some minor issues with the manuscript are separately noted.

Specific comments

1. The paper needs to clearly specify its purpose in the abstract and its purpose and

scope in the introduction. As it currently is, it is clear that the paper is about the VIACS AB, but why the paper is needed and what specifically it will cover is not described.

2. Lines 140-141: Australia is a long standing region of VIA research which is not contained within the regions noted.

3. Lines 194-196: The text notes here that one way in which Climate Services are distinguished from meteorological forecast services is the 'probabilistic nature of most of the climate information'. This isn't really quite right. Weather forecasts can be probabilistic, and climate information need not be. I think the main thing is that the uncertainties around climate projections are much larger than in weather forecasting, and as a consequence need a different approach.

4. Para at lines 212-221: Description given of figure 2a, the current situation in communication between VIACS and modelling communities. I think this is accurate in general, but in some countries national projection services provide more coordinated lines of communication (e.g. UK, Australia) to VIACS communities (but not back to CMIP). This should be noted.

5. Lines 265-270: Constitution of the Board. How are members appointed?

6. Line 410 – 561: Description of impact sector communities: These seem somewhat uneven, differing more than the nature of the communities would require. Some specific examples in the next two points.

7. Lines 465-480: Water Resources: This sector stood out as looking poorly organised compared to the others. Is that really the case?

8. Lines 536-554: Terrestrial ecosystems: This description in unbalanced in its focus on the US situation as opposed to that elsewhere. Also 'climate services' are referred to in this item, but not the previous ones. Is there are real distinction being made between sectors with regard to climate services?

9. Line 700: Recommendations for new data sets. Do the authors mean 'few recommendations' (as written), or 'a few recommendations'? A number are given, so the latter may be better.

Minor issues

Lines 57-58: 'sell-being' should be 'well-being'

Line 676: 'worth provision of' is an odd expression.

---

## Short Comment (SC1) · 1 Jun 2016

The CMIP Panel is undertaking a review of the CMIP6 GMD special issue papers to ensure a level of consistency among the invited contributions, also in answering the key questions that were outlined in our request to submit a paper to all co-chairs of CMIP6-Endorsed MIPs. We very much welcome the important contribution from the VIACS AB to CMIP6, and below are a few comments:

- Please consistently use the term 'CMIP6-Endorsed MIPs' when you refer to other MIPs that are endorsed by CMIP6 (e.g. in line 39, 118, 282, 761) - Please ensure consistency of the experiment short name and other abbreviations with the CMIP6 overview paper (see Eyring et al., 2016) (e.g. line 117: please replace with "Diagnostic, Evaluation and Characterization of Klima (DECK) experiments (klima is Greek for

"climate")". - Please ensure consistency with the final abbreviations and full names of the CMIP6-Endosed MIPs (see Table 3 of Eyring et al., 2016) (e.g. DynVarMIP instead of DynVar; long name 'Dynamics and Variability Model Intercomparison Project' / 'for CMIP6' removed in long name of the VIACS AB in our Table 3)). - Section 5.3: Some server side calculations are envisaged to provide output on common grids. Please could you specify the list of variables for which such regridding would be most helpful for the VIACS community? - Table 1: Could you please replace 'Central Set' with 'Entry card simulations for CMIP6'? - Table 3: Please specify for which experiments the variables are requested. While Key Message 4 on page 19 includes a list of experiments that are of interest to the VIACS community, the experiments should additionally be specified here. In particular, are the variables requested from all DECK experiments and the CMIP6 historical simulations or only the latter? If in addition these variables are requested for a subset of the CMIP6-Endosed MIPs, please specify these experiments as well. Are you committing to analyze all the data that you are requesting? - Line 119: Please change 'drive individual experiments' to 'define individual experiments' since the modelling groups run the simulations, not the CMIP6-Endorsed MIPs themselves. - Line 121ff: please update the paragraph on the WCRP Grand Science Challenges (see Eyring et al., 2016) - Line 202: we suggest adding one more bullet to this list: the definition of variables for the CMIP6 data request that are relevant for the VIACS community. Ensuring the relevant output is included in the CMIP6 data request is a prerequisite for any analysis, so we see this is as a major need for this communication.

References: - Eyring, V., Bony, S., Meehl, G. A., Senior, C. A., Stevens, B., Stouffer, R. J., and Taylor, K. E.: Overview of the Coupled Model Intercomparison Project Phase 6 (CMIP6) experimental design and organization, Geosci. Model Dev., 9, 1937-1958, doi:10.5194/gmd-9-1937-2016, 2016.

With many thanks for your ongoing efforts in the CMIP6 process.

The CMIP Panel

---

## Author Comment (AC1) · 28 Jun 2016

We wish to express our thanks to the anonymous referees and interactive commenter for their detailed and constructive comments on "The Vulnerability, Impacts, Adaptation, and Climate Services (VIACS) Advisory Board for CMIP6" (by A.C. Ruane and co-authors; Geosci. Model Dev. Discuss., doi:10.5194/gmd-2016-71, 2016). Below please find our responses to referee #1 below each comment (beginning with "Authors' Response:"), which detail the resulting changes we made to the manuscript given tight space constraints. We believe that the manuscript is substantially improved as a result.

Best regards,

-Alex Ruane and co-authors

[Figure]

===================================================================

Comments from Anonymous Referee #1:

+ The paper looks great and it covers various issues on the application of CMIP6 coupled models for studies on VIACS. However, CMIP6 models do not include impact models, and without that it would be hard to perform VIA applications. I am happy that the PROVIA be part of this, but as far as I can see vulnerable regions such as SE Asia, Central and South America and Africa (countries other than South Africa) are out of the Advisory Board, as is shown in the paper. May by PROVIA can cover this gap but PROVIA is not the advisory board.

Authors' Response: The VIACS Advisory Board members were drawn overwhelmingly from existing projects and international programs, which are disproportionately led by North American and European leadership (although the Board also includes representatives from South Africa on both the VIA and CS side). This disproportionate representation is also a reflection of discrepancies in the VIA publications (as noted in Section 2.2.1). Unfortunately there were few regions that have organized anything like a VIACS community with consolidated points of contact and leadership, so the regional aspect of engagement proved more difficult. The lack of representation from East Asia, Latin America, and Oceania is an acknowledged shortcoming. We now include a brief discussion of our challenges in identifying regional representatives and state in the text that we will seek to better balance out regional representation in the next iteration of the Board (Section 3.3). We have also tried to overcome regional limitations through our participation with PROVIA, the Climate Services Partnership, and the networks cultivated within each of the projects and programs (many of which include leadership from these under-served regions).

+ The authors wrote that "CMIP6 provided a unique opportunity to facilitate a two-way dialogue between CMIP6 climate modelers and VIACS". However, I can not see how this dialogue would come up in the paper. In my experience, climate modelers work

not so much on societal relevant issues, and the simulation outputs data not always are ready to use for impacts studies. The VIA community sometimes is not familiar with model outputs. So, from what I see in the paper, the problems may continue, no matter how complex would be CMIP6 models.

Authors' Response: We agree that the VIACS Advisory Board alone is not enough to eliminate differing interests, priorities, and expertise, and have added to the Summary and Benefits (Section 6) to highlight these continuing issues (some of which are healthy) The VIACS Advisory Board process does highlight a shared interest in breaking down these expectations and barriers in terms of the scope of interest, as we have found that the VIACS community is motivated to better understand the models and their output, while the modeling community has a profound interest in seeing their work used for societally-relevant applications. We have added a to the "Visualizations, Documentation, and Guidance" Section (5.4) to call more explicitly for joint efforts on translating climate model outputs into vetted, bias-corrected, accessible, and usefully-formatted VIA inputs.

+ Around line 290, "Representatives of the VIACS Advisory Board also participate in major CMIP6 meetings to give voice to the VIACS perspective", what kind of participation? what kind of voice? It is the voice of the members of the VIA community that may not be part of this board? How the VIACS board works with modellers?

Authors' Response: We have added to this section (3.4) to highlight the role of VIACS Advisory Board members at CMIP meetings (most recently at the Workshop on CMIP5 Model Analysis and Scientific Plans for CMIP6 in Dubrovnik). At CMIP6 meetings the VIACS Advisory Board representative acts as a resource when the climate modelers have questions about likely interests or ramifications of decisions on the VIACS community, may suggest actions and frameworks that facilitate VIACS research, and promotes engagement between the two communities through the VIACS Advisory Board process. Any formal recommendation by the VIACS Advisory Board must be discussed across the wider Advisory Board as no single member may make Board recommenda-

tions. Members of the VIACS Advisory Board may also interact with climate modelers through a wide variety of tasks related to their non-Advisory-Board work, but the process by which the VIACS Advisory Board formally works with the modeling community is through CMIP6 leadership. This process is outlined in section 3.4 of the manuscript and summarized in Table 2.

+ Working on VIA myself, I realized that not everything is solved by climate coupled models, as those from CMIP5 or CMIP6, nor by regional climate models. Regional climate models and experiments such as CORDEX and others represent model applications that may generate data that would feed impacts models.

Authors' Response: Agreed; we have included the RCM community within the VIACS Advisory Board because of their vital role in producing input information for VIA models. We explicitly call out the importance of CORDEX in Section 3.2 to underscore this cross-scale need. CORDEX also exists as a diagnostic MIP within CMIP as its contributions go beyond the VIACS orientation, but CORDEX leadership provides valuable perspective as to how to engage and build communications in both the VIACS and climate modeling realms. We have added to Section 3.2 to call out the importance of CORDEX and TGICA (which plays a similar role for scenarios).

+ The paper describes some sectors in which the VIA focus would be, but the authors would not say how this will be done.

Authors' Response: We have added a reminder of the engagement process within the VIACS Advisory Board consultation steps (summarized in Table 2) in the introductory paragraph of the section (#4) describing the various sectors and communities.

+ There should be a section on impact models and how uncertainties would be assessed.

Authors' Response: This is clearly an important issue and one that is cross-cutting across all VIACS sectors, but recommendations on how uncertainties are assessed

is beyond the scope of this paper. We have added the Board's interest in facilitating a common approach to assessing the uncertainty cascade from climate into VIACS models and assessments as the final recommendation within Section 5.4 (Visualizations, Documentation, and Guidance).

+ It is my suggestion that this paper's focus should be wider than the advisory committee, and it should reflect a global reality, by adding authors from regions such as India, China, Central or Southern Africa, Central and South America, so the global flavour would be on it. It would be nice that the advisory committee on VIACS be more regionally representative, but at least that paper must reflect the different realities, and I suggest including authors from vulnerable regions.

Authors' Response: As discussed in the first refereed comment above, we have modified the text to better express our shared interest in a more inclusive and representative VIACS Advisory Board (Section 3.3). We have also noted that many Board members work in regions beyond our home countries, which provides some limited perspective even as the need for more inclusive membership in future Boards remains.

---

## Author Comment (AC2) · 28 Jun 2016

We wish to express our thanks to the anonymous referees and interactive commenter for their detailed and constructive comments on "The Vulnerability, Impacts, Adaptation, and Climate Services (VIACS) Advisory Board for CMIP6" (by A.C. Ruane and co-authors; Geosci. Model Dev. Discuss., doi:10.5194/gmd-2016-71, 2016). Below please find our responses to referee #2 below each comment (beginning with "Authors' Response:"), which detail the resulting changes we made to the manuscript given tight space constraints. We believe that the manuscript is substantially improved as a result.

Best regards,

-Alex Ruane and co-authors

[Figure]

===============================================================

Comments from Anonymous Referee #2:

This is paper is useful, informative, and well written. I appreciate the work of the authors in putting it together.

The paper is particularly successful at providing an overview of important communities engaged in work at the nexus between climate and VIA research, and in shedding light on recent activities of the VIACS AB in facilitating communication between these different communities.

A few thoughts on how this draft might be improved:

***1) I'm not sure that either the abstract or the introduction provides an accurate map of the paper?

I would have expected some early part of this paper to say something like "This paper describes the motivation that led to the development of the VIACS AB, provides an overview of the various communities it attempts to engages, and summarizes recent activities." Or something similar. For this reader, it wasn't entirely clear where the paper was going until the end.

Authors' Response: We have revised the abstract to better reflect the overall text and are grateful for the specific suggestion.

***2) I'm also curious if this paper could be a bit more ambitious in offering a vision for the VIACS AB?

I see this paper as saying that the VIACS AB will facilitate communication between disparate communities, and then summarizing some recent activities to that end. This is fine, but I'm wondering if the paper couldn't go a bit further in synthesizing what sorts of information / messages / lessons the VIACS AB has learned from different kinds of communities? And can the paper identify some major questions that need to

be resolved or addressed by VIA researchers engaging climate modelers?

Pulling this out of section 4, where the state of work in various communities is described, and out of section 5 (particularly key messages from the prioritization activity) would provide a sense of key issues that this group will need to tackle and a greater perspective on the orientation of the co-chairs. It would also offer a more compelling conclusion, offering a bridge between the summary section and the benefits.

Authors' Response: The manuscript touches on these questions in several sections, most notably the motivation for the VIACS Advisory Board (Section 3.1), the section describing VIACS Activities to date (Section 5; and especially the key messages from the Prioritization of CMIP experiments and outputs – Section 5.1), and the benefits to various communities listed in the Summary and Benefits (Section 6). At this stage the VIACS community has not performed any formal effort to capture and synthesize questions for the VIACS community from the climate modeling community (or vice versa), relying instead on the initial questions described within Section 5 as these proved most pressing in the design of CMIP6. We have added a note in Section 5.4 to indicate that the VIACS Advisory Board would be interested in a formal survey of interests, lessons, and messages, which could be an interesting area of future work; however the Board exists to communicate these messages (should they be developed by PROVIA, CMIP climate modelers, or the Climate Services Partnership) rather than to conduct this type of survey. We have adapted Section 5.4 to include a discussion on future work, which we believe more tightly wraps up Sections 4 and 5 and leaves the reader with a better sense of where the Board is going. We feel the Summary and Benefits (Section 6) is still a useful closing section as the aim of this manuscript is really to describe the motivation, creation and mandate of the VIACS Advisory Board, with compelling initial results serving to demonstrate its potential but not superseding the establishment of the Board itself.

+ Though it's coming a bit off the cuff, I'm also wondering if there's some way to link that kind of synthesis to the three science questions of CMIP, or their VIA interpretation

on p 7?

Authors' Response: We have moved the VIACS interpretation of CMIP's three science questions into the summary section, as this is a more appropriate section to revisit these topics. This is particularly true following our development of Section 5.4 into a forward looking section that touches on several of the key science elements (uncertainty, scenarios, bias correction, etc.).

***3) On a related topic, I'm wondering if there's scope to propose future activities for the AB?

I see that the paper suggests establishing a formal link with the GFCS, and that the conclusion section indicates that the VIACS AB will be most successful if it identifies contact points and networks that allow for a broad and inclusive interaction. It may also be that section 5.4 is describing future, rather than present / past, actions.

But I'm wondering if there's something more that can be said? Are the authors able to articulate some priority actions that would give readers a clearer sense of what they see as most important steps? In many cases, this may just be a matter of distilling material that appears earlier, a bit less directly, into the conclusion section.

From my perspective, this kind of distillation would provide readers with a more concrete sense of what the board plans to do, and an easier read.

Authors' Response: Thanks to suggestions from all reviewers, we have developed Section 5.4 to explicitly call out some future activities, including uncertainty assessment, bias correction, scenario generation, cross-cutting engagement, visualization, and the identification and transfer of best practices utilizing the combined expertise of the climate modeling and climate applications communities.

***4) Will there be a follow up paper that addresses how / whether CMIP addressed the guidance it got from the AB?

The key messages section is really interesting : : : I'd be interested as well to hear

what actions were taken in response to the advice provided. Is any of that available now?

Authors' Response: In the period between submission and revision the CMIP team has developed specific model output packages in response to the VIACS requests. The data archive may therefore now be searched by users to request specific variable sets requested by VIACS communities. We now mention this responsive action prominently at the bottom of Section 5.1, in the summary (section 6), and in the data availability section.

+ Addressing a few of these issues would force the authors to synthesize things a bit more, and to offer perhaps a more elaborated view of the role they see the VIACS AB playing in the future. I think this would add value to the paper and provide the reader a better sense of the board members' vision.

Authors' Response: Agreed (see actions taken in responses above).

---

## Author Comment (AC3) · 28 Jun 2016

We wish to express our thanks to the anonymous referees and interactive commenter for their detailed and constructive comments on "The Vulnerability, Impacts, Adaptation, and Climate Services (VIACS) Advisory Board for CMIP6" (by A.C. Ruane and co-authors; Geosci. Model Dev. Discuss., doi:10.5194/gmd-2016-71, 2016). Below please find our responses to referee #3 below each comment (beginning with "Authors' Response:"), which detail the resulting changes we made to the manuscript given tight space constraints. We believe that the manuscript is substantially improved as a result.

Best regards,

-Alex Ruane and co-authors

[Figure]

======================================================================

Comments from Anonymous Referee #3:

General comments

The VIACS Advisory Board is an excellent initiative. It addresses a key gap between climate modellers and the user community, particularly in the context of the CMIP ensembles.

This paper describes the VIACS AB, including how it is constituted, its scope, mode of operation, objectives etc. The paper does this well and is a worthy publication, requiring little in the way of significant modification. Following here are some specific comments for the authors and readers to consider. I note that some of these may apply more to the VIACS AB operation in general than to this manuscript in particular. Some minor issues with the manuscript are separately noted.

Specific comments

1. The paper needs to clearly specify its purpose in the abstract and its purpose and scope in the introduction. As it currently is, it is clear that the paper is about the VIACS AB, but why the paper is needed and what specifically it will cover is not described.

Authors' Response: This was also recommended by Reviewer #2, and we have accordingly provided a stronger statement about the scope and purpose of this paper to lead off the abstract. We have also improved the coherence of the message from the abstract through the introduction, and provided a future work component of Section 5.4 that we believe better wraps up the initial results and findings before the summary and benefits section.

2. Lines 140-141: Australia is a long standing region of VIA research which is not contained within the regions noted.

Authors' Response: We have added Asian-Pacific to the list and have also augmented
the VIACS AB Structure (Section 3.3) to provide more information about the state of regional representation (as per comments from Reviewer #1 above).

3. Lines 194-196: The text notes here that one way in which Climate Services are distinguished from meteorological forecast services is the 'probabilistic nature of most of the climate information'. This isn't really quite right. Weather forecasts can be probabilistic, and climate information need not be. I think the main thing is that the uncertainties around climate projections are much larger than in weather forecasting, and as a consequence need a different approach.

Authors' Response: Our thanks to the reviewer for pointing out this error. The probabilistic nature is indeed inherent in weather forecasts as well. We removed this sentence since more detailed exploration of differences between weather forecasting and climate projections is an unnecessary tangent in this section. Instead we kept only the multidisciplinary nature of the information required (which also highlights this point more).

4. Para at lines 212-221: Description given of figure 2a, the current situation in communication between VIACS and modelling communities. I think this is accurate in general, but in some countries national projection services provide more coordinated lines of communication (e.g. UK, Australia) to VIACS communities (but not back to CMIP). This should be noted.

Authors' Response: We now include the role of national projection services as an additional line of communication in Section 3.1 to better represent communications in some countries within the text around Figure 2a.

5. Lines 265-270: Constitution of the Board. How are members appointed?

Authors' Response: We have now added to Section 3.3 (Structure of the VIACS Advisory Board) to indicate the origin of the original co-Chairs and selection of Board Members (leaders in their sectors, international programs, or major projects). Members were appointed by the co-chairs, with heavy consultation among leaders of the various communities. The preliminary members of the Advisory Board arose from joint discussions in the lead up to CMIP6 wherein various communities noted parallel efforts to organize and consolidate communications within the VIACS community and between the VIACS and Climate Modeling communities.

6. Line 410 – 561: Description of impact sector communities: These seem somewhat uneven, differing more than the nature of the communities would require. Some specific examples in the next two points.

Authors' Response: Each co-author re-examined their sections in an effort to more closely harmonize description frameworks across all VIACS communities. Particular attention was given to the specific sections highlighted below, however additional information was also provided for CORDEX and PROVIA.

7. Lines 465-480: Water Resources: This sector stood out as looking poorly organized compared to the others. Is that really the case?

Authors' Response: The sector is probably less well organized than the others. There is little coordination between catchment-scale studies, and the global-scale research community is small – but increasingly coordinated. The text has been revised to make this more explicit, and also to add a few more specific details.

8. Lines 536-554: Terrestrial ecosystems: This description in unbalanced in its focus on the US situation as opposed to that elsewhere. Also 'climate services' are referred to in this item, but not the previous ones. Is there are real distinction being made between sectors with regard to climate services?

Authors' Response: This section has been updated such that the use examples are used to indicate some of the efforts which are also being considered internationally, especially with agricultural efforts in the UN through PROVIA and biodiversity efforts of IPBES assessments and analysis.

9. Line 700: Recommendations for new data sets. Do the authors mean 'few recommendations' (as written), or 'a few recommendations'? A number are given, so the latter may be better.

Authors' Response: We have revised the text to clarify that it is "only a few recommendations", as we wish to emphasize that it is not a large number (compared to those contributed by other MIPs).

Minor issues

Lines 57-58: 'sell-being' should be 'well-being'

Authors' Response: Corrected

Line 676: 'worth provision of' is an odd expression.

Authors' Response: Agreed; and revised.

————————————————

---

## Author Comment (AC4) · 28 Jun 2016

We wish to express our thanks to the anonymous referees and interactive commenter for their detailed and constructive comments on "The Vulnerability, Impacts, Adaptation, and Climate Services (VIACS) Advisory Board for CMIP6" (by A.C. Ruane and co-authors; Geosci. Model Dev. Discuss., doi:10.5194/gmd-2016-71, 2016). Below please find our responses to the short comment from the CMIP Panel below each specific comment (beginning with "Authors' Response:"), which detail the resulting changes we made to the manuscript given tight space constraints. We believe that the manuscript is substantially improved as a result.

Best regards,

-Alex Ruane and co-authors

[Figure]

==================================================================

Short Comment from the CMIP Panel:

The CMIP Panel is undertaking a review of the CMIP6 GMD special issue papers to ensure a level of consistency among the invited contributions, also in answering the key questions that were outlined in our request to submit a paper to all co-chairs of CMIP6-Endorsed MIPs. We very much welcome the important contribution from the VIACS AB to CMIP6, and below are a few comments:

- Please consistently use the term 'CMIP6-Endorsed MIPs' when you refer to other MIPs that are endorsed by CMIP6 (e.g. in line 39, 118, 282, 761)

Authors' Response: Corrected

- Please ensure consistency of the experiment short name and other abbreviations with the CMIP6 overview paper (see Eyring et al., 2016) (e.g. line 117: please replace with "Diagnostic, Evaluation and Characterization of Klima (DECK) experiments (klima is Greek for "climate")".

Authors' Response: We have made corrections and double-checked with Eyring et al., 2016, for consistency.

- Please ensure consistency with the final abbreviations and full names of the CMIP6-Endosed MIPs (see Table 3 of Eyring et al., 2016) (e.g. DynVarMIP instead of DynVar; long name 'Dynamics and Variability Model Intercomparison Project' / 'for CMIP6' removed in long name of the VIACS AB in our Table 3)).

Authors' Response: We have made corrections and double-checked with Eyring et al., 2016, for consistency.

- Section 5.3: Some server side calculations are envisaged to provide output on common grids. Please could you specify the list of variables for which such regridding would be most helpful for the VIACS community?

[Figure]

Authors' Response: We have added to Section 5.3 to indicate that preliminary regridding would be most useful for monthly temperatures, precipitation, solar radiation, and humidity.

- Table 1: Could you please replace 'Central Set' with 'Entry card simulations for CMIP6'?

Authors' Response: Replaced

- Table 3: Please specify for which experiments the variables are requested. While Key Message 4 on page 19 includes a list of experiments that are of interest to the VIACS community, the experiments should additionally be specified here. In particular, are the variables requested from all DECK experiments and the CMIP6 historical simulations or only the latter? If in addition these variables are requested for a subset of the CMIP6-Endorsed MIPs, please specify these experiments as well.

Authors' Response: As we now indicate in the Section 5.1 (Key Message 2), specific experiments for which new variables were requested varied across VIACS groups, but they were most often requested for the Historical, DECK, and ScenarioMIP experiments. These variables were also requested for 12 of the 17 CMIP6-Endorsed MIPS. New variables are grouped by sector in Table 3, which now also indicates which CMIP6-Endorsed MIPs had experiments that were specifically requested. Note that we have also added variables for the Energy Sector (which were submitted in recent weeks).

- Are you committing to analyze all the data that you are requesting?

Authors' Response: As we indicate in the text of Key Message 2 within Section 5.1, the VIACS Advisory Board does not analyze data itself, but the communities that requested the data through the VIACS Advisory Board have indicated a commitment to analyze those data.

- Line 119: Please change 'drive individual experiments' to 'define individual experiments' since the modelling groups run the simulations, not the CMIP6-Endorsed MIPs

themselves.

Authors' Response: Changed

- Line 121ff: please update the paragraph on the WCRP Grand Science Challenges (see Eyring et al., 2016)

Authors' Response: Updated and simplified with reference to Eyring et al., 2016

- Line 202: we suggest adding one more bullet to this list: the definition of variables for the CMIP6 data request that are relevant for the VIACS community. Ensuring the relevant output is included in the CMIP6 data request is a prerequisite for any analysis, so we see this is as a major need for this communication.

Authors' Response: Agreed; we have added this bullet.

+ References:

- Eyring, V., Bony, S., Meehl, G. A., Senior, C. A., Stevens, B., Stouffer, R. J., and Taylor, K. E.: Overview of the Coupled Model Intercomparison Project Phase 6 (CMIP6) experimental design and organization, Geosci. Model Dev., 9, 1937-1958, doi:10.5194/gmd-9-1937-2016, 2016.

With many thanks for your ongoing efforts in the CMIP6 process.

The CMIP Panel

Authors' Response: We appreciate the efforts of the CMIP Panel to include the VIACS Advisory Board and provide feedback on this manuscript.

---

## Author Response (AR2)

August 5th, 2016

Dear GMD Editors,

5 Attached please find our updated manuscript, "The Vulnerability, Impacts, Adaptation and Climate Services Advisory Board (VIACS AB v1.0) contribution to CMIP6". We thank you and the anonymous reviewer for your time and effort on this manuscript. As shown in the tracked changes version below, we have adjusted the title as requested and have made some other minor adjustments to update the text (references, several clarifications, and a bit more information

10 about the energy impacts sector).

Please let us know if any further information is required at this time, and we look forward to working with you to complete the review and publication process.

15 Best regards,

Alex Ruane (on behalf of all co-authors)
alexander.c.ruane@nasa.gov

[revised manuscript text omitted]